# Assessing the Effectiveness of Artificial Intelligence Models for Detecting Alveolar Bone Loss in Periodontal Disease: A Panoramic Radiograph Study

**DOI:** 10.3390/diagnostics13101800

**Published:** 2023-05-19

**Authors:** Bilge Cansu Uzun Saylan, Oğuzhan Baydar, Esra Yeşilova, Sevda Kurt Bayrakdar, Elif Bilgir, İbrahim Şevki Bayrakdar, Özer Çelik, Kaan Orhan

**Affiliations:** 1Department of Periodontology, Faculty of Dentistry, Dokuz Eylul University, İzmir 35220, Turkey; 2Department of Oral and Maxillofacial Radiology, Faculty of Dentistry, Ege University, İzmir 35040, Turkey; 3Department of Dentomaxillofacial Radiology, Faculty of Dentistry, Eskişehir Osmangazi University, Eskişehir 26040, Turkey; 4Department of Periodontology, Faculty of Dentistry, Eskişehir Osmangazi University, Eskişehir 26040, Turkey; 5Department of Mathematics and Computer Science, Faculty of Science, Eskisehir Osmangazi University, Eskisehir 26480, Turkey; 6Department of Oral and Maxillofacial Radiology, Faculty of Dentistry, Ankara University, Ankara 06830, Turkey

**Keywords:** alveolar bone loss, artificial intelligence, panoramic radiography, deep learning, segmentation

## Abstract

The assessment of alveolar bone loss, a crucial element of the periodontium, plays a vital role in the diagnosis of periodontitis and the prognosis of the disease. In dentistry, artificial intelligence (AI) applications have demonstrated practical and efficient diagnostic capabilities, leveraging machine learning and cognitive problem-solving functions that mimic human abilities. This study aims to evaluate the effectiveness of AI models in identifying alveolar bone loss as present or absent across different regions. To achieve this goal, alveolar bone loss models were generated using the PyTorch-based YOLO-v5 model implemented via CranioCatch software, detecting periodontal bone loss areas and labeling them using the segmentation method on 685 panoramic radiographs. Besides general evaluation, models were grouped according to subregions (incisors, canines, premolars, and molars) to provide a targeted evaluation. Our findings reveal that the lowest sensitivity and F1 score values were associated with total alveolar bone loss, while the highest values were observed in the maxillary incisor region. It shows that artificial intelligence has a high potential in analytical studies evaluating periodontal bone loss situations. Considering the limited amount of data, it is predicted that this success will increase with the provision of machine learning by using a more comprehensive data set in further studies.

## 1. Introduction

Due to its inherited characteristics, radiology has always been a science that transfers the developments in physics and mathematics to medical imaging and has the capacity to renew itself, ever since the discovery of X-rays [1]. Given the harmful biological effects, creating images with minimum exposure to X-rays, using radiofrequency and ultrasound to examine soft tissues, and developing digital imaging systems are essential steps in radiology [2,3]. Radiological techniques have been widely used in clinical dentistry practice since 1896; teeth and alveolar bone can be clearly visualized radiographically due to the inorganic components in their structures [1]. 

Periodontal diseases are characterized by pathological alterations that occur in the periodontium, which surrounds the tooth, comprising the gingival tissue, alveolar bone, cementum, and periodontal ligaments. Periodontitis is a chronic, multifactorial, inflammatory disease associated with the accumulation of dental plaque and characterized by the progressive destruction of the tissues supporting the teeth, including the periodontal ligament, and alveolar bone [4,5]. It can cause irreversible bone resorption, tooth mobility, and tooth loss if not adequately treated. At the same time, periodontal diseases have the potential to predispose individuals to various systemic diseases such as cardiovascular disease, oral and colorectal cancer, gastrointestinal diseases, respiratory tract infection and pneumonia, adverse pregnancy outcomes, diabetes and insulin resistance, and Alzheimer’s disease [6]. Although early diagnosis is essential in managing periodontal disease, determining the level of periodontal disease also affects the treatment protocols to be applied. 

Despite advances in treatment modalities, new methods are needed to detect alveolar bone loss in more detail and to evaluate bone loss in problematic teeth. Panoramic, periapical, and bite-wing radiographs are widely used to diagnose periodontally problematic teeth and predict pathologies [7]. Panoramic radiography is an imaging technique for displaying all teeth and jaws with surrounding tissues. The overall evaluation of teeth and related bone loss can be comprehensively defined by panoramic radiography. However, this technique has some disadvantages for practitioners to overcome to establish a correct diagnosis. The two-dimensional image of three-dimensional and curved jaws cannot always be displayed on the screen under ideal conditions. Even though this results in incomplete evaluation of patients [8,9], this technique provides a comprehensive assessment of alveolar bone level with less radiation dose compared to 3D modalities.

Artificial intelligence (AI) represents the learning and problem-solving capacity of machines that mimic humans’ cognitive functions [10,11,12]. The most important development in artificial intelligence is the provision of machine learning. This is based on creating algorithms that can learn from data and make predictions based on that data. The first step of machine learning is preparing a training dataset with sufficient data and labeling it appropriately, which is the most crucial step. The tagged data is saved in the appropriate format and pre-processed to ensure the proper training steps. Training is carried out using 2D and 3D convolutional neural network (CNN) architectures, by giving various training steps. In a simpler language, large quantities of data are introduced by drawing the boundaries of the structures to be taught to the system, and it is ensured that the system automatically learns this and converts it into output through multi-layered artificial neural networks [13].

AI was first used in dental radiology by White as ORAD (https://www.orad.org/cgi-bin/orad/index.pl, accessed on 9 October 2022) in 1995. This method gave assistance to practitioners for other possible diagnoses in radiography. One of the main reasons researchers aim to incorporate AI into radiology practice is its capacity to extract data from large numbers of location-inconsistent studies and to identify imaging markers that can predict treatment outcomes or responses. Neural network analysis of large datasets can reveal important associations that could never be detected by visual interpretation of studies alone and may prove important in future personalized healthcare. Considering that these relationships are products of mathematical algorithms, important results can be obtained from these data mining outputs. The key is that AI has the potential to enable the integration of data mining of electronic medical records in the process, as well as replacing many of the routine detection, characterization, and quantification processes currently practiced by radiologists using cognitive ability [14,15,16].

The idea that visual diagnosis can be improved using artificial intelligence in radiology to produce lower error rates than the human observer has ushered in an exciting era with clinical and research capabilities. Detection and classification of lesions, automatic image segmentation, data analysis, extraction of radiographic features, and converting these into an automatic printout are important technological developments for computer-aided medicine applications, in medical and especially radiological terms [14,15].

In recent studies in dentistry, AI models have been developed for many pathologies and anatomies and each study has tried to increase the number of successful models. Studies are aimed at to common problems in dentistry, i.e., caries lesions, periodontal diseases, endodontic lesions, and jawbone pathologies [17,18,19].

There exist studies based on CNNs which utilize not only two-dimensional (2D) images but also three-dimensional (3D) images and have achieved notable success [20,21]. Although three-dimensional (3D) evaluations are widely used in various fields of dentistry, including implantology, surgery, endodontics, and orthodontics, their application in periodontology is primarily limited to the assessment of furcations, craters, and bone defects, as well as the determination of root form and alveolar relationship [22]. During a standard periodontal evaluation, periapical, bite-wing, and panoramic radiography are the preferred methods for assessing the level of alveolar bone in the interproximal area. This is due to the cost-effectiveness, rapidity, and lower radiation exposure associated with these 2D imaging techniques when compared to 3D imaging modalities [23]. In consideration of these factors, we opted to employ panoramic radiography images, which are commonly utilized in periodontal assessments for evaluating alveolar bone loss, in our study.

The aim of this study is to evaluate the success of artificial intelligence models used in the detection of radiographic alveolar bone loss using segmentation method in different regions of the jaw by orthopantomography images.

## 2. Materials and Methods

### 2.1. Study Design

In this study, alveolar bone loss by region models were determined in panoramic radiographs using a Pytorch-implemented YOLO-v5 model. Eskisehir Osmangazi University Non-interventional Clinical Research Ethics Board (decision date and decision number: 04.10.2022/22) approved the study protocol. The principles of the Helsinki Declaration were followed in the study.

### 2.2. Data Sources

In the study, an intraoral examination was not performed, and radiology data were evaluated retrospectively. Only radiographic evaluation was made and areas compatible with bone loss findings were labeled with the segmentation method. To establish the study dataset, orthopantomography images acquired in January 2022 from the archive of Eskişehir Osmangazi University Faculty of Dentistry were scanned. Adult patients were included in the study, and the dose and duration of radiography were the same as the standard panoramic radiography procedure and process in all patients. Images with many metal artifacts, incorrect patient positioning, low quality due to patient movement, rare bone morphologies, orthognathic treatment, and those in which the affected area could not be accurately selected for periodontal bone destruction determination were excluded from the study. Age, gender, and ethnicity differences were not observed, and the data was anonymized prior to uploading into the labeling system. All images were acquired utilizing a standardized panoramic imaging device (Planmeca Promax 2D Panoramic System, Planmeca, Helsinki, Finland) with acquisition parameters consisting of 68 kVp, 14 mA, 12 s, 15 μSv, and a pixel size of 48 μm. In this study, 1543 panoramic radiographs were randomly selected and uploaded as a project. Afterward, 685 of these radiographs were labeled and alveolar bone loss labeling was also applied to all radiographs.

### 2.3. Labeling and Training of Data

Labeling is the process of identifying areas in an image and determining which region the object belongs to. The labeling of the images was performed using the CranioCatch (CranioCatch, Eskişehir, Turkey) labeling module. The evaluators (oral and maxillofacial radiology and A periodontology specialist with at least 10 years of experience) agreed on how to conduct the labeling process according to the criteria they had previously determined, and the labeling was performed by the periodontologist in consultation with the radiologist. For labeling, alveolar bone losses in 8 different regions were evaluated. Regions were classified as incisors, canines, premolars, and molars for the maxilla and mandible. At the same time, general alveolar bone loss was also identified as a separate label, regardless of region. The presence and absence of alveolar bone loss was recorded considering the distance between the cemento-enamel junction of the teeth (CEJ), approximately 2 mm apical, and the alveolar crest (AC). The bone loss’s outer borders were determined by polygonal segmentation and saved in JSON (JavaScript Object Notation) format.

For the segmentation model, 685 anonymized, mixed-sized panoramic radiography images were resized to 1280 × 512 diameter. A random dataset was created by using the open-source Python programming language and OpenCV, NumPy, Pandas, and Matplotlib libraries. In order to prevent the use of the images participating in the training for re-testing, the data set was divided into three parts: 80% training, 10% validation, and 10% testing.
Training group: 80% of the images (represents the data set used for training the model.)Validation group: 10% of the images (indicates the samples that are independent of the training of the model and should not be seen by the model during this period. The model is tested on this dataset to stop training or revise training variables.)Test group: 10% of the images (constitutes the data set in which the trained model is tested by using the training and validation data.)

The training, testing, and validation dataset numbers for each evaluated parameter are presented in Table 1. Training and validation datasets were used to predict and generate optimal AI models. The success of the model was checked with the test data set. The dataset was trained using transfer learning with a pre-trained model.

### 2.4. Deep-Learning Algorithm

Classified and labeled images were resized to 1280 × 512 pixels in the training. The dataset was trained using transfer learning with a pre-trained model. In this study, the training was carried out with the PyTorch library in Python (v. 3.6.1: Python Software Foundation, Wilmington, DE, USA), using 2D and 3D CNN architectures, over 500 training epochs. YOLO-v5 was used for segmentation training. YOLO-v5 is a model in the family of computer vision models. There are four main versions of YOLO-v5, each offering progressively higher accuracy rates: small (s), medium (m), large (l), and extra-large (x). We used the YOLO-v5x model here. The YOLO-v5x architecture consists of the same 3 components, backbone, neck, and head. CSPDarknet53 is used as backbone. On the neck, path aggregation network (PANet) and spatial pyramid pooling (SPP) are used. The data are first input to CSPDarknet for feature extraction and then fed to PANet for feature fusion. Finally, YOLO Layer outputs segmentation results (class, score, location) [24,25,26].

In the YOLO-v5x model, sigmoid linear unit (SiLU) (non-linear activation function) was used as the activation function. SiLU also performs the function that helps reduce overfitting. In this study, the SGD optimizer was used, we set the learning rate parameter of this optimizer as 0.01 and the momentum parameter as 0.937 in default settings. We used the early stop method to avoid overfitting. Epoch 500 is used as batch-size 16 and image size 1280 × 512. The total number of layers may vary depending on different versions of YOLO-v5. For example, YOLO-v5s has 86 layers, YOLO-v5m has 170 layers, and the YOLO-v5x model we used has 238 layers. We used the batch normalization technique for the YOLO-v5 model. Batch normalization is a normalization technique used to increase learning speed and performance at every layer of the network. We used the SGD (stochastic gradient descent) method as the gradient descent method. The loss function used in the YOLO-v5x model is focal loss, specially customized for multi-class object detection. Focal loss is specifically designed to improve the poor classification performance of classes with small samples [24,25,26].

Cross-validation and hold-out testing of the YOLO-v5 model starts with dividing the dataset into equal parts. Each segment divided is used as the test set in turn, while the remainder is used as the training set. This process measures the performance of the model for each piece of the dataset separately. Both methods are important for measuring the overall performance of the model and help prevent overfitting. Here, the early stop method is also used to prevent overfitting. For example: if the model training gets the best parameters at the 36th epoch, the training will continue for another 100 epochs and automatically stop itself. The best pattern was recorded in the 36th epoch [24,25,26].

In this study, no fine-tuning was made, and we completed the training of properly labeled data in the default settings of YOLO-v5 (lr0: 0.01, lrf: 0.01, momentum: 0.937, weight_decay: 0.0005, warmup_epochs: 3.0, warmup_momentum: 0.8, warmup_bias_lr: 0.1). As a result, we tested the pytorch model output on 10% test data set and reached the accuracy metrics. The test data used to evaluate the success of the artificial intelligence model were evaluated by the radiologist (EB) and accepted as the gold standard in determining the sensitivity values.

### 2.5. Statistical Analysis

The performance of the model was evaluated using a confusion matrix. This matrix illustrates a comparison of the predicted and actual situations. In order to evaluate the success of the model, performance metrics were calculated using true positive (TP: the number of labels which labeled of the bone loss area correctly), true negative (TN: the number of cases which detected of the area without bone loss correctly), false positive (FP: the number of labels which labeled bone loss area even though there was no alveolar bone loss) and false negative (FN: the number of bone loss areas not labeled) metrics to evaluate the success of the model. These metrics were sensitivity (TP/(TP + FN)), precision (TP/(TP + FP)), and the F1-score, which is the harmonic medium of these two metrics, respectively. Sensitivity represents the proportion of actual positive cases that the model has correctly identified as positive. Precision is another metric that measures how accurate the results are. It indicates how many of the examples positively classified by the model are truly positive. In other words, in the case of a disease or dental condition, precision shows how accurately the model has classified the disease examples as positive. Precision is calculated only on the positively classified examples. F1 score is the harmonic mean of precision and sensitivity. It takes into account both precision and sensitivity and is a way to balance the trade-off between them. A high F1 score means that both precision and sensitivity are high, which indicates a model with a good balance between detecting true positives and avoiding false positives.

## 3. Results

The sensitivity, precision, and F1 score values are presented obtained for alveolar bone loss in Table 2. Additionally, the estimates of alveolar bone loss and training results of the model are illustrated in Figure 1, Figure 2, Figure 3, Figure 4, Figure 5, Figure 6, Figure 7, Figure 8 and Figure 9. The sensitivity, precision, and F1 score values for alveolar bone loss, maxillary incisor ABL, maxillary canine ABL, maxillary premolar ABL, maxillary molar ABL, mandibular incisor ABL, mandibular canine ABL, mandibular premolar ABL, and mandibular molar ABL were found as 0.75–0.76–0.76, 1–0.90–0.95, 0.88–0.81–0.84, 0.94–0.69–0.80, 0.87–0.96–0.91, 0.83–0.89–0.86, 0.92–0.78–0.84, 0.83–0.75 –0.79, and 0.85–0.73–0.79, respectively (Table 2), (Figure 1, Figure 2, Figure 3, Figure 4, Figure 5, Figure 6, Figure 7, Figure 8 and Figure 9). (In the images section the term ‘real’ indicates the labeled data, and ‘prediction’ indicates the prediction of the AI model.) Based on these results, despite the largest number of labeled images, overall alveolar bone loss appeared to have a lower F1 score compared to site-specific estimates. Considering the regions, the highest F1 score was obtained in the maxillary incisor region. When examining the jaws, it is seen that artificial intelligence models are more successful in determining alveolar bone loss in the upper jaw. 

## 4. Discussion

The study aimed to evaluate the success of artificial intelligence models in detecting alveolar bone loss by different alveolar regions. It was determined that local alveolar bone loss detection was more successful than total alveolar bone loss detection in all regions. 

In the medical field, the number of studies aiming to evaluate anatomical and pathological structures with artificial intelligence has increased recently [27,28,29]. If we look at the main problems that play a role in the development of artificial intelligence applications in dentistry, the possibility of human-induced erroneous diagnoses due to reasons such as insufficient number of experienced physicians, limited time for radiographic interpretation by physicians, and the reporting requirement of radiographs pose a problem in terms of time, cost and patient care. As the use of artificial intelligence in dentistry becomes more widespread and comprehensive programming methods are developed, these problems may gradually decrease over time. In addition, by using artificial intelligence programs that shorten the diagnosis process and provide more reliable results, the diagnosis of dental restorations, maxillofacial abnormalities, dental deformities, and periodontal and endodontic lesions from panoramic radiographs becomes more practical [30,31].

Panoramic radiography is recommended as a standard protocol for comprehensive diagnosis and treatment planning. Panoramic radiography offers the advantage to clinicians of imaging the teeth and surrounding bone with low-dose radiation [32]. The use of these images obtained with low-dose radiation in clinical practice has always found a place because the application of the technique is easily tolerated by patients and this image, which provides valuable information in the clinic, can be obtained in a short time [33]. Two-dimensional radiological examination using periapical radiography is still considered the standard method for assessing marginal bone loss. In addition, decay, root morphology, and resorptions can be identified [34]. Panoramic radiographs may occasionally be combined with periapical radiographs as an alternative to a full-mouth series of periapical radiographs to reduce the total radiation dose [33].

During a routine oral examination, it is essential that dentists can fully diagnose periodontal disease of all patients. The identification and diagnosis of periodontal disease not only guides the selection of appropriate treatment protocols but also influences their implementation and execution. In line with this objective, radiographs, which are commonly employed by clinicians, play a crucial role in diagnosing and devising an effective treatment plan for patients [35]. Periapical radiographs and periodontal probing are widely used radiographic and clinical methods in periodontal examinations [36]. To prevent tooth loss in periodontal diseases, the amount of supporting tissues of the tooth and especially the amount of bone surrounding the tooth has great importance [37]. New methods used in radiological findings, especially the average percentage values in bone loss, are of great importance for the early diagnosis and timing of treatment planning of periodontal diseases [38]. The amount of alveolar bone destruction displayed in routine panoramic radiographs is essential for the prognosis and diagnosis of periodontal diseases [39]. Alveolar bone loss is one of the main parameters used to determine the stage, complexity, prevalence, and distribution of periodontal disease, according to the new classification of periodontal and peri-implant diseases published in 2017 [34]. The development of automated systems for periodontal disease classification according to clinical and radiological features has been investigated since 1987. However, alveolar bone loss, which is very important in the detection of periodontal diseases, is a very up-to-date approach, especially in AI studies [40,41]. Additionally, the results of studies evaluating the ability of AI to detect periodontal bone loss vary depending on imaging techniques, the quantity of data, and the algorithms used. In this study, an artificial intelligence program was developed that can be used to detect total and regional periodontal bone loss on panoramic radiographs. The successful results obtained in the present study support the idea that more comprehensive artificial intelligence models could be developed for the new periodontology classification.

In the literature, artificial intelligence studies have taken their place for solving problems with the help of digital-based systems in many dental conditions. Moran et al., constructed data sets containing interproximal region images obtained from periapical radiographs and tested the performance of CNN models. The study indicated high accuracy and predictive values, and it was reported that the ResNet and Inception models performed well in the radiological evaluation of the periodontium and were a diagnostic tool that could be used in bone destruction [42]. In this study, the success of artificial intelligence with ROC and PR curves was tested using 467 periapical radiographic data. In our study, the performance of the artificial intelligence model was evaluated with the F1 value using the confusion matrix.

Lee et al. investigated the performance of CNNs in the detection and classification of periodontally compromised teeth based on alveolar bone loss. According to the results of their studies, it has been shown that the Deep CNN algorithm can diagnose periodontally compromised teeth with an accuracy close to that of a periodontologist, and it has been reported that deep learning-based CNNs can be useful in this regard [43]. A similar study using cropped panoramic radiography pictures and a trained CNN found that the CNN performed similarly to dentists in diagnosing periodontal bone loss in the incisor, premolar, and molar regions, suggesting that machine learning could improve diagnosis [44]. Although the literature studies are up-to-date and few in number, it is obvious that artificial intelligence models are promising in determining the periodontal radiographic condition. In the future, it will be possible to develop more detailed diagnostic methods with artificial intelligence that are not physician-oriented.

Jiang et al. evaluated the overall and localization-specific effectiveness of the deep learning architecture by labeling reference points that would indicate alveolar bone loss by three periodontologists on 640 panoramic radiography images [9]. It was reported that the accuracy, precision, and F1 score values of a two-level deep learning model using UNet and YOLO-v4 algorithms were generally 0.77. In different alveolar regions, these ratios were in the range of 0.71–0.81, 0.72–0.80, and 0.71–0.81, respectively. The research results show that, in parallel with the results of our study, deep learning algorithms can be used to evaluate periodontal bone loss in teeth in different regions. Although the same quantity of data was used in our study, the F1 score and sensitivity values were found to be slightly higher. We used YOLO-v5, which is significantly faster than YOLO-v4, in our study.

Thanathornwong et al. evaluated the detectability of periodontally compromised teeth using faster regional CNNs using one hundred panoramic radiographs [45]. The sensitivity, precision, and F1 score values of the model used were 0.84, 0.81, and 0.81, respectively, indicating that the algorithm can accelerate the detection of periodontally compromised teeth. Chang et al. used deep learning methods in panoramic radiography images to properly categorize diseases based on bone loss, and high accuracy and reliability rates were obtained with the framework they established on computer-aided diagnosis [46]. In our study, we observed that the lowest sensitivity and F1 score values were associated with total alveolar bone loss, while the highest values were in the maxillary incisor region.

Bayrakdar et al. evaluated the performance of CNNs to detect radiographs with alveolar bone loss using the GoogleNet architecture using 2276 panoramic radiography images [47]. They used the classification method and the sensitivity, precision, and F1 score values of CNN were 0.94, 0.89, and 0.91, respectively. These results show that dentists can facilitate the diagnosis and treatment planning of bone loss. On the other hand, Lee et al., in another study, aimed to measure the performance of deep learning to show alveolar bone loss and level by using intraoral radiography images. It was shown that the deep learning algorithm detects the absence of bone loss with a sensitivity of 0.96, and the sensitivity ranges between 0.80 and 0.93 according to the stages of bone loss [41]. In a different study, DeNTNet, a deep learning-based technique that uses panoramic radiography pictures to automatically detect alveolar bone loss, was used. It has been stated that DeNTNet has a higher precision and recall score for periodontal loss compared to dentists, and thus can save clinicians speed and time in dental applications [48]. We assume that this difference in outcomes in panoramic radiography investigations is due to the technique being more sensitive to patient movements and positions than intraoral imaging modalities and intraoral radiographs providing better resolution than panoramic radiographs. Variations in the success of panoramic radiography studies may stem from differences in the quantity of data used in the study, as well as the preferred architectures during the development of the artificial intelligence model. Additionally, some studies may utilize distinct techniques, such as classification, object detection, or segmentation, to identify diseased areas and conduct image processing. Therefore, it is not feasible to make direct comparisons between success rates obtained from various studies. 

The limitations of this study were the lack of intraoral probing and ideal gold standard. Similar to the retrospective radiology-based studies in the literature, alveolar bone loss was determined only by radiography evaluation and 2 mm apical of the CEJ was accepted as the reference point in the radiography [49,50,51]. In order to overcome this limitation in future studies, there is a need for new studies that can evaluate bone destruction from radiography, including cone beam computed tomography (CBCT) and intraoral clinical measurement. We had 685 panoramic radiographs scanned retrospectively, showing segmental alveolar bone loss. In future studies, more detailed indicators and high success rates will be obtained with the increase in the quantity of data to be processed on panoramic radiography.

Better results are obtained in the anterior region of the upper jaw; the maxilla is more spongy bone, and the borders can be determined more quickly depending on the radiopacity contrast difference between the alveolar bone and the tooth. In particular, the cortical structure of the lower jaw can cause difficulty in determining the border between the tooth and the alveolar bone. However, the smaller and tighter positioning of the anterior teeth in the lower jaw compared to the upper jaw is also one of the factors affecting the measurements. 

## 5. Conclusions

Artificial intelligence has recently reached an interesting position in the medical field. An increasing number of studies are conducted with the idea that artificial intelligence applications in dentistry can serve as a clinical decision-support mechanism and help physicians like a second eye. If the percentage of periodontal bone loss can be determined with artificial intelligence in future studies, it will contribute to the new periodontal classification. In this study, our first goal was to label alveolar bone destruction on alveolar bone radiography with AI and show it to the physician using the segmentation method.

In our study, it was concluded that regional detection was more successful than general detection in panoramic radiography of periodontal bone loss. It is thought that with the use of this method in panoramic radiography, dentists will be able to detect more effectively and easy regional periodontal bone loss in the future.

## Figures and Tables

**Figure 1 diagnostics-13-01800-f001:**
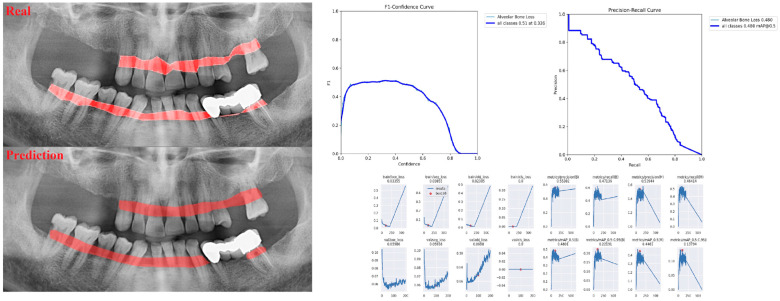
Prediction of alveolar bone loss with AI in panoramic images and training results.

**Figure 2 diagnostics-13-01800-f002:**
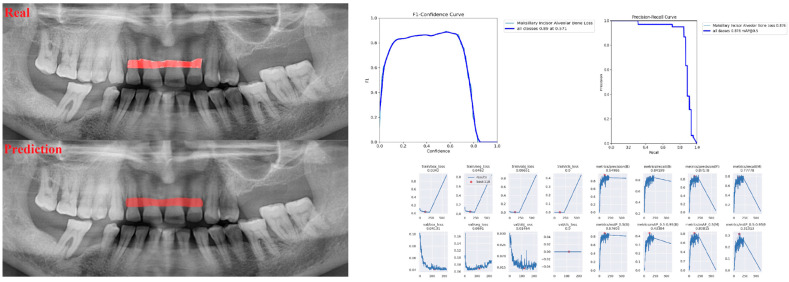
Prediction of alveolar bone loss of maxillary incisor region with AI in panoramic images and training results.

**Figure 3 diagnostics-13-01800-f003:**
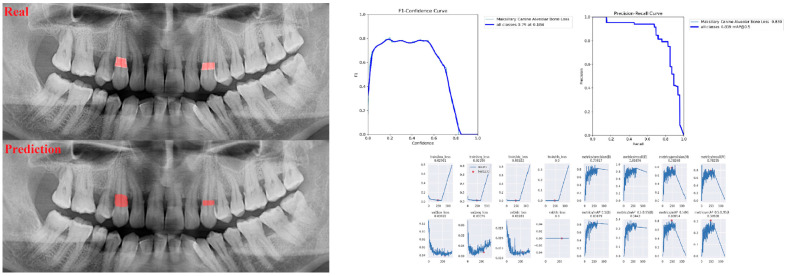
Prediction of alveolar bone loss of maxillary canine region with AI in panoramic images and training results.

**Figure 4 diagnostics-13-01800-f004:**
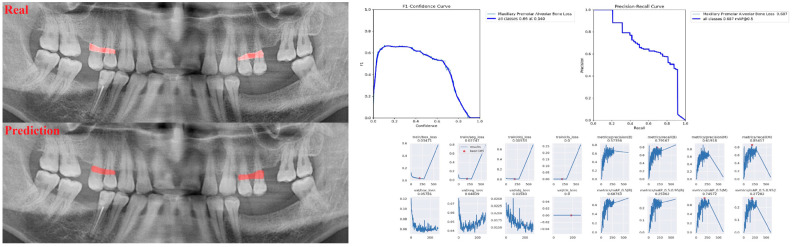
Prediction of alveolar bone loss of maxillary premolar region with AI in panoramic images and train results.

**Figure 5 diagnostics-13-01800-f005:**
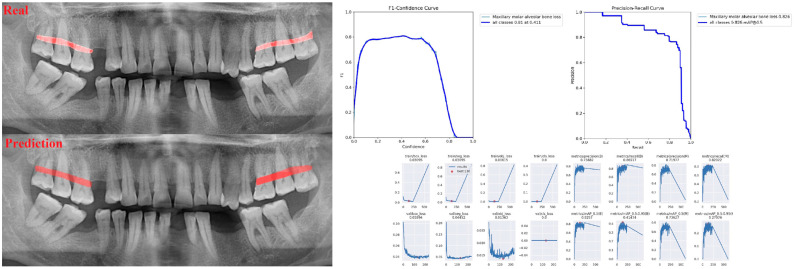
Prediction of alveolar bone loss of maxillary molar region with AI in panoramic images and training results.

**Figure 6 diagnostics-13-01800-f006:**
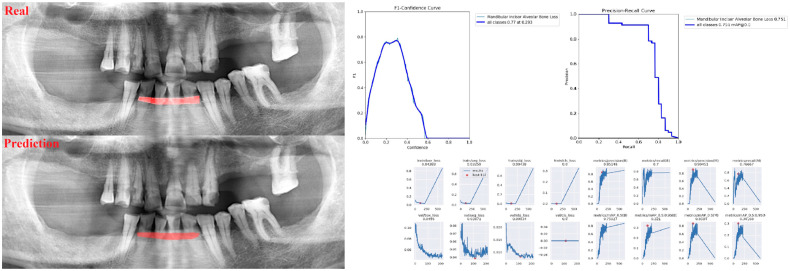
Prediction of alveolar bone loss of mandibular incisor region with AI in panoramic images and training results.

**Figure 7 diagnostics-13-01800-f007:**
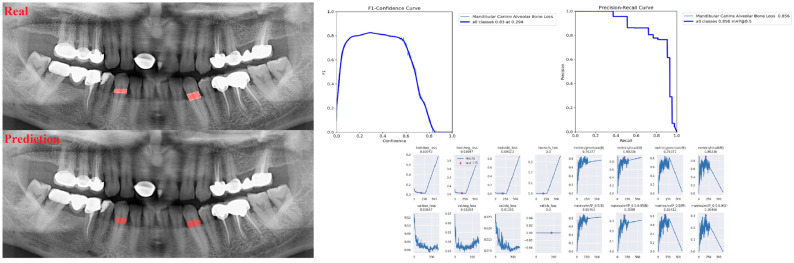
Prediction of alveolar bone loss of mandibular canine region with AI in panoramic images and training results.

**Figure 8 diagnostics-13-01800-f008:**
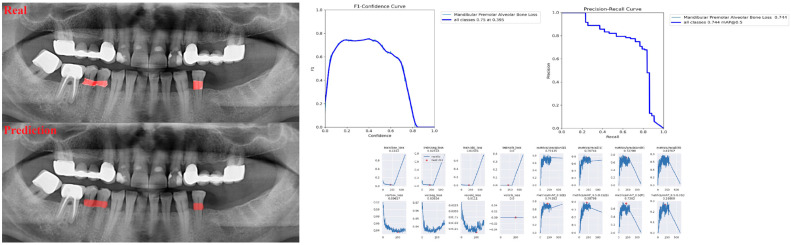
Prediction of alveolar bone loss of mandibular premolar region with AI in panoramic images and training results.

**Figure 9 diagnostics-13-01800-f009:**
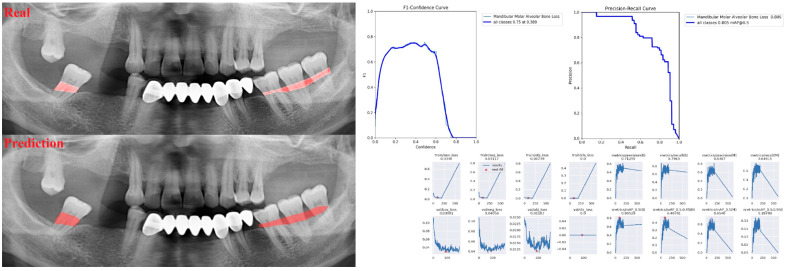
Prediction of alveolar bone loss of mandibular molar region with AI in panoramic images and training results.

**Table 1 diagnostics-13-01800-t001:** Distribution of the data in the study according to training, testing, and validation.

Panoramic Images			
Diagnoses	Training Group	Testing Group	Validation Group
Alveolar Bone Loss (ABL)	549	68	68
Maxillary			
Incisor ABL	352	43	44
Canine ABL	253	31	32
Premolar ABL	284	35	35
Molar ABL	445	55	56
Mandibular			
Incisor ABL	238	29	30
Canine ABL	214	26	27
Premolar ABL	233	29	29
Molar ABL	277	34	35

**Table 2 diagnostics-13-01800-t002:** The sensitivity, precision, and F1 score value of AI model estimation performance measures in general and different regions.

Measurement Value			
	Sensitivity	Precision	F1 Score
Alveolar Bone Loss (ABL)	0.75	0.76	0.76
Maxillary			
Incisor ABL	1	0.90	0.95
Canine ABL	0.88	0.81	0.84
Premolar ABL	0.94	0.69	0.80
Molar ABL	0.87	0.96	0.91
Mandibular			
Incisor ABL	0.83	0.89	0.86
Canine ABL	0.92	0.78	0.84
Premolar ABL	0.83	0.75	0.79
Molar ABL	0.85	0.73	0.79

## Data Availability

Open access to the data was not considered due to legal and ethical issues in the state.

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
