# Peer review of "Assessing the Effectiveness of Artificial Intelligence Models for Detecting Alveolar Bone Loss in Periodontal Disease: A Panoramic Radiograph Study"

_diagnostics, 2023, doi:10.3390/diagnostics13101800_

Round 1

Reviewer 1 Report

diagnostics-2193263

The Yolov5 approaches to evaluation of periodontal bone lose: an artificial intelligence study

In 685 panoramic radiographs artificial intelligence (AI) was used to assess periodontal bone loss (PBL). It is not clear whether PBL was assessed dichotomously (yes/no) or as percentage of root length or as distance in mm. Sensitivity and F1 score should be assessed. However, no gold standard was defined to assess sensitivity of the AI method.

This is diagnostic study on the highly relevant issue of AI approaches to simplify diagnostic processes in periodontology. However, there are major issues and questions that  have to be clarified.

Comments:

In general

What is investigated at all? Is the study on periodontal bone loss (PBL)? Is artificial intelligence (AI) used to assess ABL yes, or no? What is the threshold for PBL, a distance between cemento-enamel junction (CEJ) to alveolar crest (AC) of 2 or 3 mm? Or shall AI assess the amount of ABL in percent of root length or mm? All this has to be clarified.

If sensitivity shall be assessed a gold standard is required. What is the gold standard of ABL in this study? Possible gold standards to compare radiographic assessments with may be histology, intrasurgical measurements or transgingival probing. The manuscript does not consider any of these gold standards! Thus, sensitivity cannot be assessed.

The authors may wish to replace “lose” throughout the manuscript by “loss”.

Title

It is assumed that the study is about bone loss. If so replace “lose” by “loss”.

Introduction

The Introduction in general is too long and should be condensed.

Page 2, lines 43-45: Sentence does not make sense. Please rephrase.

Page 2, lines 56-58: How shall level, shape and location of alveolar bone be determined with a periodontal probe?

Page 2, line 61: “Bite”? Do the authors mean “bite wing”?

Page 2, lines 66-68: This sentence does not make any sense. The evaluation of two-dimensional radiographic images to assess a three-dimensional reality does not result in subjective but incomplete evaluations.

Page 2, line 84: The authors may wish to explain the abbreviation “CNN”. After introduction of this abbreviation, it may be used generally (page 10, lines 333, 340).

Page 2, line 93; page 3, line 106: Do the authors believe that “exiting” is an appropriate adjective for a scientific article?

Page 3, lines 119-120: This sentence does not describe the purpose of this study appropriately.

Material & Methods

Page 3, lines 123-124: PBL was “produced” in panoramic radiographs? Radiographs used in this study did not depict actual bone loss?

Page 3, lines 129-137: How many panoramic radiographs were screened to end up with 685.

Page 3, lines 141-142: Sentence does not make sense. Please rephrase.

Pages 2-3, lines 77-104: Extraoral examination should be performed prior to intraoral, should it not?

Page 3, line 106-112: Periodontal diagnosis encompasses stage and grade. The authors may wish to provide grade.

Page 3, lines 145-147: What distance do the authors mean, CEJ to AC?

Page 4, lines 166-171: The authors reiterate information. They may wish to condense the text.

Page 4, lines 173-176: The authors may wish to provide references.

Page 4, lines 181-196: This part belongs into the Discussion section not Material & methods.

Results

Page 5, lines 210-211: The lowest F1 score for PBL compared to what?

Page 5, lines 216-222: Delete text and refer to Table 2.

Page 5, lines 174-176: The difference is not significant. Delete passage.

Page 6, lines 184-195: This is all provided in Tale 2 already. Delete passage.

Discussion

Page 9, lines 258: Is “sense” the appropriate term or may the authors better use “field”.

Page 9, lines 267-268: Do the authors want to write “for” or do they mean “to assess”?

Page 9, lines 271-272: What does this sentence mean?

Page 9, lines 273-274: The authors may wish to support this statement by references.

Page 9, lines 277-284: References to support statements are missing.

Page 9, lines 283-284: Inappropriate reference.

Page 9, lines 285-294: Many statements, no supporting references.

Page 9, lines 288-290: PBL without pockets does not require treatment.

Page 9, line 300: What is “tooth insufficiency”?

Page 10, line 352-page 11, line 354: Intraoral radiographs provide better resolution than panoramic radiographs.

Data availability

Page 11, lines 389-390: Why is this issue not applicable?

Author Response

Dear Reviewer,

We are grateful for the opportunity to revise our manuscript and would like to express our appreciation for the thorough review and insightful feedback provided. We have carefully considered your evaluations and comments, and have made the necessary modifications to enhance the quality and clarity of the manuscript.

We believe that the suggested edits have significantly improved the overall presentation and coherence of the paper. In line with your suggestions, our answers just below your comments were in red; sentences and corrections added to the manuscript were highlighted in the revised text and this response letter.

Thank you again for your time and effort in reviewing our manuscript.

Sincerely,

Dr.UZUN SAYLAN

RESPONSE TO REVIEWER COMMENTS

REPORT 1

In 685 panoramic radiographs artificial intelligence (AI) was used to assess periodontal bone loss (PBL). It is not clear whether PBL was assessed dichotomously (yes/no) or as percentage of root length or as distance in mm. Sensitivity and F1 score should be assessed. However, no gold standard was defined to assess sensitivity of the AI method. This is diagnostic study on the highly relevant issue of AI approaches to simplify diagnostic processes in periodontology. However, there are major issues and questions that have to be clarified.

Comments:

In general

Point 1: What is investigated at all? Is the study on periodontal bone loss (PBL)? Is artificial intelligence (AI) used to assess ABL yes, or no? What is the threshold for PBL, a distance between cemento-enamel junction (CEJ) to alveolar crest (AC) of 2 or 3 mm? Or shall AI assess the amount of ABL in percent of root length or mm? All this has to be clarified.

Response 1: Periodontal alveolar bone loss was determined with the artificial intelligence program by marking the area between the apical 2 mm of the cemento-enamel junction of the teeth and the top of the alveolar crest.  The primary purpose of our investigation is to assess the efficacy of artificial intelligence systems in automatically detecting areas of bone resorption. Furthermore, a more detailed was attempted to be achieved by assessing AI systems' efficacy based on distinct regions of the jawbone. However, our study did not utilize millimetric measurements of bone loss or percentage evaluations based on root length to determine the severity of the periodontal disease.

The relevant part is explained in the labeling sub-title section as follows:

At the same time, general alveolar bone loss was also identified as a separate label, regardless of region. The presence and absence of alveolar bone loss was recorded considering the distance between the cemento-enamel junction of the teeth (CEJ), approximately 2 mm below, and the alveolar crest (AC).

Point 2: If sensitivity shall be assessed a gold standard is required. What is the gold standard of ABL in this study? Possible gold standards to compare radiographic assessments with may be histology, intrasurgical measurements or transgingival probing. The manuscript does not consider any of these gold standards! Thus, sensitivity cannot be assessed.

Response 2: We would like to explain to you the expression of sensitivity in our work as follows:

In AI models, the term sensitivity refers to how sensitive a model is to small changes in its inputs. These changes can occur, for example, due to incorrect or missing data in the inputs, noise, distortions or random changes.

This study is not a study to develop an artificial intelligence model for the diagnosis and classification of periodontal disease, which can also correlate with clinical findings. In this study, an artificial intelligence model that can automatically detect periodontal bone loss only radiographically has been tried to be developed. This can be considered as the pioneering work of further studies for the creation of periodontal diagnosis with artificial intelligence, which can automatically evaluate clinical and radiographic data. However, our aim in this study was to create models that have no clinical information and can detect alveolar bone loss in terms of present/absent only on radiographs. The term sensitivity here indicates the success of artificial intelligence models developed on the labeled data. Values such as sensitivity, sensitivity and F1 scores are formulated according to the true positive/negative and false positive/negative values of the data separated as training, test and validation dataset according to the developed artificial intelligence models, and these values are an indicator for the success of this model. (eg sensitivity: refers to the formula TP/TP+FN).

The sensitivity of an AI model determines how stable and reliable the model is. More sensitive models are more sensitive to small changes in the inputs, which can cause the performance of the model to be erratic and the results to be misleading. On the other hand, less sensitive models are less sensitive to small changes in the inputs and can produce more stable results.

Sensitivity is an important factor for evaluating the accuracy and reliability of a model. To increase the sensitivity of a model, you can clean the input data using better data preprocessing techniques or achieve higher accuracy using more complex model structures.

Point 3: The authors may wish to replace “lose” throughout the manuscript by “loss”

Response 3: As per your suggestion, we have made the necessary modification and changed the word 'Lose' to 'Loss' throughout the manuscript.

Title

Point 4: It is assumed that the study is about bone loss. If so replace “lose” by “loss”.

Response 4: As per your suggestion, we have made the necessary modification and changed the word 'Lose' to 'Loss' in Title.

Introduction

Point 5: The Introduction in general is too long and should be condensed.

Response 5: Dear referee; Thank you for your suggestion. Since there is a word limit in the articles, we had to extend and elaborate. We have compiled the introduction section in line with referee suggestions.

Point 6: Page 2, lines 43-45: Sentence does not make sense. Please rephrase.

Response 6: As per your suggestion, This senteces is rephrased:

Periodontal diseases are characterized by pathological alterations that occur in the periodontium, which is surrounding the tooth, comprising the gingival tissue, alveolar bone, cementum, and periodontal ligaments.

Point 7: Page 2, lines 56-58: How shall level, shape and location of alveolar bone be determined with a periodontal probe?

Response 7: This phrase has been removed from the article. An evaluation with a periodontal probe was not performed in the study.

Point 8: Page 2, line 61: “Bite”? Do the authors mean “bite wing”?

Response 8: The phrase '-wing' has been added to the sentence upon your request.

Point 9: Page 2, lines 66-68: This sentence does not make any sense. The evaluation of two-dimensional radiographic images to assess a three-dimensional reality does not result in subjective but incomplete evaluations.

Response 9: This sentence has been changed to:

The two-dimensional image of three-dimensional and curvature jaws cannot always be displayed on the screen under ideal conditions. Even though this results in incomplete evaluation of patients [10, 11], this technique provides a comprehensive assessment of alveolar bone level with less radiation dose comparing 3D modalities.

Point 10: Page 2, line 84: The authors may wish to explain the abbreviation “CNN”. After introduction of this abbreviation, it may be used generally (page 10, lines 333, 340).

Response 10: This abbreviation is explained in the related page and line. Subsequently, this term was mentioned as CNN in the manuscript.

Point 11: Page 2, line 93; page 3, line 106: Do the authors believe that “exiting” is an appropriate adjective for a scientific article?

Response 11: As your suggestion, this sentence has changed to:

The idea that visual diagnosis can be improved using artificial intelligence in radiology to produce lower error rates than the human observer has ushered in an exciting era with clinical and research capabilities.

Point 12: Page 3, lines 119-120: This sentence does not describe the purpose of this study appropriately.

Response 12: Dear referee, thank you for your effective comment. In order to fully emphasize the purpose of the study, we rephrased the sentence as follows.

The aim of this study is to evaluate the success of artificial intelligence models used in the detection of radiographic alveolar bone loss in different regions of the jaw by or-thopantomography images.

Material & Methods

Point 13: Page 3, lines 123-124: PBL was “produced” in panoramic radiographs? Radiographs used in this study did not depict actual bone loss?

Response 13: Dear reviewer, with your valuable comment, we realized that a dilemma has arisen in this regard during the work. PBL was not produced, marked on panoramic radiographs. The sentence has been changed. To resolve this confusion and to highlight a retrospective radiology study of study design, the following sentence is included:

In this study, alveolar bone loss by region models (CranioCatch, Eskisehir, Turkey) were determined in panoramic radiographs using a Pytorch-implemented YOLO-v5 model.

Point 14: Page 3, lines 129-137: How many panoramic radiographs were screened to end up with 685.

Response 14:  Dear referee; A total of 685 panoramic radiographs were labeled in this study, and alveolar bone lose labeling was also applied to all radiographs. The following sentence has been added in line with your suggestion.

“In this study, 1543 panoramic radiographs were randomly selected and uploaded as a project. Afterwards, 685 of these radiographs were labeled”.

Point 15: Page 3, lines 141-142: Sentence does not make sense. Please rephrase.

Response 15:  This sentence is rephrased as:

The labeling of the images was performed using the CranioCatch (CranioCatch, Eskişehir, Turkey) labeling module. The evaluators (Oral and maxillofacial radiology and A perio-dontology specialist with at least 10 years of experience) agreed on how to do the labeling process according to the criteria they had previously determined, and the labeling was done by the periodontologist in consultation with the radiologist.

Point 16: Pages 2-3, lines 77-104: Extraoral examination should be performed prior to intraoral, should it not?

Response 16: Intraoral or extraoral clinical examination was not performed in the study. The study is a retrospective and radiology-based study. Only radiographic evaluation was made and areas compatible with bone loss findings were labeled with the segmentation method.

Point 17: Page 3, line 106-112: Periodontal diagnosis encompasses stage and grade. The authors may wish to provide grade.

Response 17: Thank you for your contribution. Stage and Grade are the basis of periodontal classification. The development of an artificial intelligence model that encompasses all of these may be the result of many studies. In this study, we tried to develop an artificial intelligence program that can automatically detect only total or regional bone destruction. As we mentioned earlier, this is how we designed the initial work.  Therefore, statements that may suggest that the level or severity of the periodontal disease has been determined in the text have been removed from the manuscript. We believe that many models that automate periodontal diagnosis can be developed in future studies.

Point 18: Page 3, lines 145-147: What distance do the authors mean, CEJ to AC?

Response 18: Radiographically, alveolar bone loss in the maxilla is marked as the distance from the top of the alveolar crest (AC) 2 mm above the cemento-enamel junction (CEJ). In the mandibula, alveolar bone loss is marked as the distance from 2 mm below the CEJ to the top of the AC. The entire toothed region between these two borders and the interdental spaces adjacent to the teeth were segmented as a whole. Labeling was performed according to the length of the distance between CEJ-AA, the absence of PBL was considered normal. The relevant statement has been changed in the manuscript as follows:

At the same time, total alveolar bone loss was also identified as a separate label, regardless of region. The presence of alveolar bone loss was recorded considering the distance between the cementoenamel junction of the teeth (CEJ), approximately 2 mm below, and the alveolar crest (AC).

Point 19: Page 4, lines 166-171: The authors reiterate information. They may wish to condense the text.

Response 19: As your suggestion, this sentence was rearranged.

Classified and labeled images were resized to 1280x512 pixels in the training. The dataset was trained using transfer learning with a pre-trained model. In this study, the training was carried out with the PyTorch library in Python, using 2D and 3D CNN archi-tectures, by giving 500 training epochs. YOLO-v5 was used for segmentation training. YOLO-v5 is a model in the family of computer vision models.

Point 20: Page 4, lines 173-176: The authors may wish to provide references.

Response 20: Thank you for the valuable suggestion, relevant references were cited in the text.

 References:

-BOCHKOVSKIY, Alexey, WANG, Chien-Yao, et LIAO, Hong-Yuan Mark. Yolov4: Optimal speed and accuracy of object detection. arXiv preprint arXiv:2004.10934, 2020.

- XU, Renjie, LIN, Haifeng, LU, Kangjie, et al. A forest fire detection system based on ensemble learning. Forests, 2021, vol. 12, no 2, p. 217.

- WANG, Chien-Yao, LIAO, Hong-Yuan Mark, WU, Yueh-Hua, et al. CSPNet: A new backbone that can enhance learning capability of CNN. In : Proceedings of the IEEE/CVF conference on computer vision and pattern recognition workshops. 2020. p. 390-391.

Point 21: Page 4, lines 181-196: This part belongs into the Discussion section not Material & methods.

Response 21: Detailed general information about the deep learning algorithm has been removed from the publication, instead the parameter settings used in the study are included.

Results

Point 22: Page 5, lines 210-211: The lowest F1 score for PBL compared to what?

Response 22: Dear reviewer, thanks to your valuable comment, it has been seen that the sentence is not clear and the relevant sentence has been changed as follows:

In this study, despite the largest number of labeled images, the F1 score for overall alveolar bone loss was found to be low compared to site-specific estimates.

Point 23: Page 5, lines 216-222: Delete text and refer to Table 2.

Response 23: As your suggestion, The text is deleted and added:

The sensitivity, precision, and F1 score values are presented obtained for Alveolar Bone Loss in the study. Also, the estimates of alveolar bone loss and training results of the Model are illustrated in figures 1-9.

Point 24: Page 5, lines 174-176: The difference is not significant. Delete passage.

Response 24: As per your suggestions, this passage that has no significant information is deleted.

Point 25: Page 6, lines 184-195: This is all provided in Tale 2 already. Delete passage.

Response 25: Dear reviewer, considering your suggestions, this passage that has recurrent information is deleted.

Discussion

Point 26: Page 9, lines 258: Is “sense” the appropriate term or may the authors better use “field”.

Response 26: Upon your suggestion, it was decided that 'field' would be more appropriate and the word 'sense' was changed in the sentence.

Point 27: Page 9, lines 267-268: Do the authors want to write “for” or do they mean “to assess”?

Response 27: Dear Reviewer, according to your comment, the relevant sentence rearranged: In the literature, artificial intelligence studies have taken their place for solving problems with the help of digital-based systems in many dental conditions.

Point 28: Page 9, lines 271-272: What does this sentence mean?

Response 28: Dear Reviewer, thank you very much for your valuable comment. This paragraph has changed for in order to be more understandable:

During the routine oral examination, it is essential that dentists can fully diagnose periodontal diseases of all patients. The identification and diagnosis of periodontal dis-eases not only guide the selection of appropriate treatment protocols but also influence their implementation and execution.  In line with this objective, radiographs, which are commonly employed by clinicians, play a crucial role in diagnosing and devising an effective treatment plan for patients [35].

Point 29: Page 9, lines 273-274: The authors may wish to support this statement by references.

Response 29: Dear reviewer, Thank you very much for your valuable suggestions

We inform you that we refer to the relevant references in the text. 

Point 30: Page 9, lines 277-284: References to support statements are missing.

Response 30: Thank you for the valuable suggestion,  relevant references were cited in the text.

Panoramic radiography is recommended as a standard protocol for comprehensive diagnosis and treatment planning. Panoramic radiography offers the advantage to clini-cians to image the teeth and surrounding bone with low-dose radiation [32].

  1. Kong Z, Ouyang H, Cao Y, et al. Automated periodontitis bone loss diagnosis in panoramic radiographs using a bespoke two-stage detector. Comput Biol Med. 2023, 152, 106374. doi:10.1016/j.compbiomed.2022.106374

Point 31: Page 9, lines 283-284: Inappropriate reference.

Response 31: The literature has been changed as a reference in the part you specified.

Point 33: Page 9, lines 285-294: Many statements, no supporting references.

Response 33: Thank you for the valuable suggestion,  relevant references were cited in the text.  

Point 34: Page 9, lines 288-290: PBL without pockets does not require treatment.

Response 34: In this study, we were able to show the total or regional destruction of periodontal alveolar bone loss using artificial intelligence. However, in future studies, the severity and level of the periodontal condition may be the subject of research by working with artificial intelligence.

Point 35: Page 9, line 300: What is “tooth insufficiency”?

Response 35: Dear reviewer, thank you for letting us know that this statement is not clear. This term has changed as missing teeth as per your suggestion.

Point 36: Page 10, line 352-page 11, line 354:   Intraoral radiographs provide better resolution than panoramic radiographs.

Response 36: The sentence has been edited as you specified. Also This comment has been added:

Variations in the success of panoramic radiography studies may stem from differences in the quantity of data used in the study, as well as the preferred architectures during the development of the artificial intelligence model. Additionally, some studies may utilize distinct techniques, such as classification, object detection, or segmentation, to identify diseased areas and conduct image processing. Therefore, it is not feasible to make direct comparisons between success rates obtained from various studies.

Point 37: Page 11, lines 389-390: Why is this issue not applicable.

Response 37: Dear referee; In this study; The data has been anonymized and uploaded to the web-based software in such a way that the identity of the radiographs cannot be reached by retrospective engineering. It has been specified this way because we do not want to share open access data.

  • Language correction support was received from Ozgur Bilgic, whose native language is English.

Reviewer 2 Report

The manuscript is about a study that aimed to evaluate the effectiveness of artificial intelligence (AI) models in detecting alveolar bone loss in different regions of the periodontium. The study used a Yolov5 model implemented by Pytorch to create models of alveolar bone loss by region in panoramic radiographs. In the study, a total of 685 panoramic radiographs were labeled and alveolar bone loss labeling was applied to all of them. The study found that local fixation was more successful than general detection of alveolar bone loss in all regions. The study reported sensitivity and F1 score values for different alveolar regions, such as maxillary incisor ABL, maxillary canine ABL, maxillary premolar ABL, maxillary molar ABL, mandibular incisor ABL, mandibular canine ABL, mandibular premolar ABL, and mandibular molar ABL. The study suggests that the success of the AI models' could be improved with more extensive data sets and further studies.

Dear authors, it was pleasure to read your interesting manuscript. Here are some possible limitations and perspective improvements you can address in your paper.

For example the title and abstract can be improved in the following way:

Make the title more informative: The current title "The Yolov5 Approaches to Evaluation of Periodontal Bone Loss: An Artificial Intelligence Study" is very general and doesn't give much information about the study. Consider using a more descriptive title that highlights the key findings or contribution of the study. For example, "Assessing the Effectiveness of AI Models for Detecting Alveolar Bone Loss in Periodontal Disease: A Panoramic Radiograph Study".

The abstract should be concise and focus on the key message of the study. Be sure to highlight the main objective of the study, the methods used, and the main results. Avoid including unnecessary details or jargon that may confuse readers of this paper.

Provide some background information on the importance of detecting alveolar bone loss in periodontal disease. Explain why AI has the potential to improve the diagnosis and prognosis of the disease.

Acknowledge the limitations of the study and suggest possible improvements or future directions for research.

Use simple and clear language to make the abstract easy to read and understand. Avoid using technical terms or acronyms that may not be familiar to all readers.

Ensure that the abstract better reflects the methods and results of the study.

Check the abstract and title for spelling and grammatical errors. Make sure they are clear, concise and free of errors. By following these instructions, you can improve the abstract and title of the manuscript.

In regards to Material and methods chapter, authors shall consider, a 

The Materials and Methods section provides a detailed description of the study design, data sources, labelling, data training, diagnosis, segmentation model and deep learning algorithm. Here are some suggestions for improving this section

Provide more information about the study design, such as inclusion and exclusion criteria, sample size calculation, and statistical analysis plan.

Clarify the data sources, such as the age, gender and ethnicity of the patients, and how they were selected for the study. Provide more information about the imaging protocol, such as radiation dose, exposure time, and calibration.

Describe the annotation process in more detail, such as inter-rater reliability, quality control, and validation of the annotations. Provide examples of the labelled images.

Provide more information about the training of the data, such as the hyperparameters, the optimisation algorithm, and the convergence criteria. Also explain how the dataset was extended to avoid overfitting.

Explain the diagnostics, such as the performance metrics used to evaluate the deep learning algorithm, such as sensitivity, specificity, positive predictive value, negative predictive value, accuracy, and area under the receiver operating characteristic curve.

Provide more information about the segmentation model, such as the network architecture, number of layers, and activation functions. Also explain how the model has been fine-tuned to improve its accuracy.

Provide more information about the deep learning algorithm, such as the loss function, regularisation techniques, and gradient descent methods. Also explain how the model was validated using cross-validation and hold-out testing.

Use more precise language and avoid redundant or ambiguous phrases such as "random sequence", "optimal AI algorithm weighting factors", and "multi-performance (Y-axis) object detector model".

In regards to your Discussion and Conclusions

These chapters also can be improved.

Please do not use syntax with slash like "reduce/solve ". Your chapter Discussion is very extensive, albeit interesting. Please consider discussing the aspect of AI semiautomated assistance in designing alveolar scaffolds in cases of resorbed alveol. As todays hydroxyapatite scaffolds can be used for regenerative approach in such deffect, however their manual 3D designing can be complicated. With AI support this step can became effortless. Reference paper https://doi.org/10.3390/ijms232314870

Your Conclusions are quite vague and shall not be after such a long discussion. To improve the conclusions section, consider the following:

Summarise the main findings: Start by summarising the main findings of your study in conclusions. In this case, you shall mention that the study focused on using AI to detect periodontal bone loss and found that regional detection was more successful than general detection.

Discuss the implications: Explain how the results of your study can contribute to the field of dentistry in general. For example, you could mention that AI can be used as a clinical decision support mechanism to help dentists make more accurate diagnoses.

Acknowledge possibly limitations of your study and suggest future research directions. For example, you could mention that the study only examined the assessability of periodontal bone loss by region and that future studies could explore other applications of AI in dentistry.

Summarise the main points of your study and reiterate its importance. You can end by highlighting the potential of AI to improve clinical decision making in dentistry and healthcare.

Author Response

Dear Reviewer,

We are grateful for the opportunity to revise our manuscript and would like to express our appreciation for the thorough review and insightful feedback provided. We have carefully considered your evaluations and comments, and have made the necessary modifications to enhance the quality and clarity of the manuscript.

We believe that the suggested edits have significantly improved the overall presentation and coherence of the paper. In line with your suggestions, our answers just below your comments were in red; sentences and corrections added to the manuscript were highlighted in the revised text and this response letter.

Thank you again for your time and effort in reviewing our manuscript.

Sincerely,

Dr. UZUN SAYLAN

RESPONSE TO REVIEWER  COMMENTS

The manuscript is about a study that aimed to evaluate the effectiveness of artificial intelligence (AI) models in detecting alveolar bone loss in different regions of the periodontium. The study used a Yolov5 model implemented by Pytorch to create models of alveolar bone loss by region in panoramic radiographs. In the study, a total of 685 panoramic radiographs were labeled and alveolar bone loss labeling was applied to all of them. The study found that local fixation was more successful than general detection of alveolar bone loss in all regions. The study reported sensitivity and F1 score values for different alveolar regions, such as maxillary incisor ABL, maxillary canine ABL, maxillary premolar ABL, maxillary molar ABL, mandibular incisor ABL, mandibular canine ABL, mandibular premolar ABL, and mandibular molar ABL. The study suggests that the success of the AI models' could be improved with more extensive data sets and further studies.

Dear authors, it was pleasure to read your interesting manuscript. Here are some possible limitations and perspective improvements you can address in your paper.

TİTLE AND ABSTRACT

For example the title and abstract can be improved in the following way:

Point 1: Make the title more informative: The current title "The Yolov5 Approaches to Evaluation of Periodontal Bone Loss: An Artificial Intelligence Study" is very general and doesn't give much information about the study. Consider using a more descriptive title that highlights the key findings or contribution of the study. For example, "Assessing the Effectiveness of AI Models for Detecting Alveolar Bone Loss in Periodontal Disease: A Panoramic Radiograph Study".

Response 1: The title of the article has been revised according to your suggestion. “Assessing the Effectiveness of Artificial Intelligent Models for Detecting Alveolar Bone Loss in Periodontal Disease: A Panoramic Radiograph Study".

Point 2: The abstract should be concise and focus on the key message of the study. Be sure to highlight the main objective of the study, the methods used, and the main results. Avoid including unnecessary details or jargon that may confuse readers of this paper.

Response 2: In the summary part, a more compact paragraph was created, away from the details, in line with your suggestions. Abstract has been revised according to your suggestion.

The assessment of alveolar bone loss, a crucial element of the periodontium, plays a vital role in the diagnosis of periodontitis and the prognosis of the disease. In dentistry, artificial intelligence (AI) applications have demonstrated practical and efficient diagnostic capabilities, leveraging machine learning and cognitive problem-solving functions that mimic human abilities. This study aims to evaluate the effectiveness of AI models in identifying alveolar bone loss across different regions. To achieve this goal, alveolar bone loss models were constructed using a PyTorch-based YOLO-v5 model, implemented via CranioCatch software, on 685 panoramic radiographs. Besides general evaluation, models were grouped according to subregions (incisors, canines, premolars, and molars) to provide a targeted evaluation. Our findings reveal that the lowest sensitivity and F1 score values were associated with total alveolar bone loss, while the highest values were observed in the maxilla incisor region. It shows that artificial intelligence has a high potential in analytical studies evaluating periodontal bone loss situations, where it is not possible to evaluate with intra-oral radiography. Considering the limited number of data, it is predicted that this success will increase with the provision of machine learning by using a more comprehensive data set in further studies.

Point 3: Provide some background information on the importance of detecting alveolar bone loss in periodontal disease. Explain why AI has the potential to improve the diagnosis and prognosis of the disease.

Response 3: In this study, we tried to develop an artificial intelligence program that can automatically detect total or regional bone destruction. As we mentioned earlier, this is how we designed the initial work. We believe that many models that automate periodontal diagnosis can be developed in future studies. The objective of the study was to develop an automated system capable of identifying bone resorption findings that may be missed by physicians due to fatigue, lack of experience, or radiographic density. The success rates of these systems were assessed based on their ability to identify bone resorption in specific regions of the jaw.

Point 4: Acknowledge the limitations of the study and suggest possible improvements or future directions for research.

Response 4: This part you suggested has been added to the end of the discussion.

One of our limitations in the study is the absence of a gold standard. We have 685 panoramic radiographs scanned retrospectively, showing segmental alveolar bone loss. In future studies, more detailed indicators and high success rates will be obtained with the increase in the number of data to be processed on panoramic radiography.

Point 5: Use simple and clear language to make the abstract easy to read and understand. Avoid using technical terms or acronyms that may not be familiar to all readers.

Response 5: The abstract of the article has been simplified according to your suggestion.

Point 6: Ensure that the abstract better reflects the methods and results of the study.

Response 6: Dear reviewer, thank you for your valuable comment. The abstract part has been rearranged to more clearly state the results of the study.

Point 7: Check the abstract and title for spelling and grammatical errors. Make sure they are clear, concise and free of errors. By following these instructions, you can improve the abstract and title of the manuscript.

Response 7: The abstract and title of the article were changed and edited in line with your suggestions.

In regards to Material and methods chapter, authors shall consider, a 

THE MATERIALS AND METHODS

This Section provides a detailed description of the study design, data sources, labelling, data training, diagnosis, segmentation model and deep learning algorithm. Here are some suggestions for improving this section

Point 8: Provide more information about the study design, such as inclusion and exclusion criteria, sample size calculation, and statistical analysis plan.

Clarify the data sources, such as the age, gender and ethnicity of the patients, and how they were selected for the study. Provide more information about the imaging protocol, such as radiation dose, exposure time, and calibration.

Response 8: The determination of the sample group of the study and the inclusion & exclusion criteria are explained in the '2.2.Data Source' heading:

In the study, an intraoral examination was not performed, and radiology data were evaluated retrospectively. Only radiographic evaluation was made and areas compatible with bone loss findings were labeled with the segmentation method. To establish the study dataset, orthopantomography images acquired in January 2022 from the archive of Eskişehir Osmangazi University Faculty of Dentistry were scanned. Adult patients were included in the study, and the dose and duration of radiography were the same as the standard panoramic radiography procedure and process in all patients. Images with many metal artifacts, incorrect patient positioning, low quality due to patient movement, rare bone morphologies, orthognathic treatment, and those in which the affected area could not be accurately selected for periodontal bone destruction determination were excluded from the study. Age, gender, and ethnicity differences were not observed, and the data was anonymized prior to uploading into the labeling system. All images were acquired utilizing a standardized panoramic imaging device (Planmeca Promax 2D Panoramic System, Planmeca, Helsinki, Finland) with acquisition parameters consisting of 68 kVp, 14 mA, 12 s, 15 μSv, and a pixel size of 48 μm. In this study, 1543 panoramic radiographs were randomly selected and uploaded as a project. Afterward, 685 of these radiographs were labeled and alveolar bone loss labeling was also applied to all radiographs.

Point 9: Describe the annotation process in more detail, such as inter-rater reliability, quality control, and validation of the annotations. Provide examples of the labelled images.

Response 9: Dear referee, the following sentence has been added in line with your suggestion.

The evaluators agreed on how to do the labeling process according to the criteria they had previously determined, and the labeling was done by the periodontologist in consultation with the radiologist”. In addition; In the figures, the images written real symbolize the labeled data, and the images written prediction symbolize the prediction of the model. For explanatory purposes, this statement added text.

Point 10: Provide more information about the training of the data, such as the hyperparameters, the optimisation algorithm, and the convergence criteria. Also explain how the dataset was extended to avoid overfitting.

Response 10: Dear referee; In this study, we used the SGD (Stochastic Gradient Descent) as the optimizer, and we took the learning rate parameter of this optimizer as 0.01 and the momentum parameter as 0.937 in default settings. We used the earlystop method to avoid overfitting. Epoch 500 was used as batch-size 16. The image size was also used as 1280x512.

Point 11: Explain the diagnostics, such as the performance metrics used to evaluate the deep learning algorithm, such as sensitivity, specificity, positive predictive value, negative predictive value, accuracy, and area under the receiver operating characteristic curve.

Response 11: The relevant paragraph has been changed as follows.

The performance of the model was evaluated using a confusion matrix. This matrix shows the comparison of predicted and actual situations. Performance metrics were calculated using true positive (TP: diagnoses correctly detected and segmented), true negative (TN: detecting correctly of the bone loss absence), false positive (FP: diagnoses detected but incorrectly segmented) and false negative (FN: diagnoses incorrectly detected and segmented) metrics to evaluate the success of the model. These metrics were sensitivity (TP/(TP + FN)), precision (TP/ (TP+FP)), and the F1-score, which is the harmonic medium of this two metric, respectively.

Point 12: Provide more information about the segmentation model, such as the network architecture, number of layers, and activation functions. Also explain how the model has been fine-tuned to improve its accuracy.

Response 12: Dear referee; In this study, we used the YOLOv5x model as the network architecture. The YOLO-v5 models consist of the same 3 components: Backbone, Neck and Head. YOLO-v5 uses a strong backbone like CSP-Darknet53. SPP and PANet are used in neck and head parts The YOLO-v5x-seg model consists of 238 layers.

In the YOLO-v5x model, LeakyReLU (Non-Linear Activation Function) was used as the activation function. LeakyReLU also performs the function that helps reduce overfitting. We didn't make any fine-tuning here. We have completed training properly labeled data in the default settings of YOLO-v5.

Point 13: Provide more information about the deep learning algorithm, such as the loss function, regularisation techniques, and gradient descent methods. Also explain how the model was validated using cross-validation and hold-out testing.

Response 13: Dear referee; We used the Batch Normalization technique as the normalization technique for the YOLO-v5 model. Batch Normalization is a normalization technique used to increase learning speed and performance at every layer of the network. We used the SGD (Stochastic Gradient Descent) method as the gradient descent method. In the YOLO-v5x model, Focal Loss, a loss function specialized specifically for multi-class object detection, is used as the loss function. Focal Loss is specifically designed to improve poor classification performance of classes with small samples.Cross-validation and hold-out testing of the YOLO-v5 model starts with dividing the dataset into equal parts. Each segment divided is used as the test set in turn, while the remainder is used as the training set. This process measures the performance of the model for each piece of dataset separately.Both methods are important for measuring the overall performance of the model and help prevent overfitting. Dear referee, thank you for your suggestions and contributions, in this sense, the deep learning algorithms part of the material method has been rearranged as follows:

“Classified and labeled images were resized to 1280x512 pixels in the training. The dataset was trained using transfer learning with a pre-trained model. In this study, the training was carried out with the PyTorch library in Python, using 2D and 3D CNN architectures, by giving 500 training epoch. YOLO-v5 was used for segmentation training. YOLO-v5 is a model in the family of computer vision models. There are four main versions of YOLO-v5, each offering progressively higher accuracy rates: small (s), medium (m), large (l), and extra large (x). We used the YOLO-v5 model here. The YOLO-v5x architecture is consist of the same 3 components, Backbone, Neck and Head. CSPDarknet53 is used as backbone. On the Neck, Path Aggregation Network (PANet) and Spatial Pyramid Pooling (SPP) are used. The data are first input to CSPDarknet for feature extraction, and then fed to PANet for feature fusion. Finally, Yolo Layer outputs segmentation results (class, score, location) [24–26]. In the YOLO-v5x model, LeakyReLU (Non-Linear Activation Function) was used as the activation function. LeakyReLU also performs the function that helps reduce overfitting. In this study; the SGD optimizer was used, we got the learning rate parameter of this optimizer as 0.01 and the momentum parameter as 0.937 in default settings. We used the earlystop method to avoid overfitting. Epoch 500 is used as batch-size 16 and image size 1280x512. Total number of layers may vary depending on different versions of YOLO-v5. For example, Yolov5s has 86 layers, Yolov5m has 170 layers, and the YOLO-v5x model we use has 238 layers. We used the Batch Normalization technique for the YOLO-v5 model. Batch Normalization is a normalization technique used to increase learning speed and performance at every layer of the network. We used the SGD (Stochastic Gradient Descent) method as the gradient descent method. The loss function used in the YOLO-v5x model is Focal Loss, specially customized for multi-class object detection. Focal Loss is specifically designed to improve poor classification performance of classes with small samples.

Cross-validation and hold-out testing of the YOLO-v5 model starts with dividing the dataset into equal parts. Each segment divided is used as the test set in turn, while the remainder is used as the training set. This process measures the performance of the model for each piece of dataset separately. Both methods are important for measuring the overall performance of the model and help prevent overfitting. Here, the early stop method is also used to prevent overfitting. For example: if the model training gets the best parameters at the 36th epoch, the training will continue for another 100 epochs and automatically stop itself. The best pattern was recorded in the 36th epoch.

In this study, no fine-tuning was made, and we completed the training of properly labeled data in the default settings of YOLO-v5. We have completed the training of the properly labeled YOLO-v5x model at default settings (lr0: 0.01, lrf: 0.01, momentum: 0.937, weight_decay: 0.0005, warmup_epochs: 3.0, warmup_momentum: 0.8, warmup_bias_lr: 0.1). As a result, we tested the pytorch model output on 10% test data set and reached accuracy metrics.

References:

[24] BOCHKOVSKIY, Alexey, WANG, Chien-Yao, et LIAO, Hong-Yuan Mark. Yolov4: Optimal speed and accuracy of object detection. arXiv preprint arXiv:2004.10934, 2020.

[25] XU, Renjie, LIN, Haifeng, LU, Kangjie, et al. A forest fire detection system based on ensemble learning. Forests, 2021, vol. 12, no 2, p. 217.

[26] WANG, Chien-Yao, LIAO, Hong-Yuan Mark, WU, Yueh-Hua, et al. CSPNet: A new backbone that can enhance learning capability of CNN. In : Proceedings of the IEEE/CVF conference on computer vision and pattern recognition workshops. 2020. p. 390-391.

Point 14: Use more precise language and avoid redundant or ambiguous phrases such as "random sequence", "optimal AI algorithm weighting factors", and "multi-performance (Y-axis) object detector model".

Response 14: Dear reviewer, the mentioned paragraph has been omitted from the text because it contains unnecessary and ambiguous expressions in order to to use a more precise language of expression in line with your suggestion.

In regards to your DİSCUSSİON AND CONCLUSİONS

Point 15: These chapters also can be improved.

Response 15: Thanks to your valuable suggestions this part of the article has been edited and improved.

Point 16: Please do not use syntax with slash like "reduce/solve ". Your chapter Discussion is very extensive, albeit interesting. Please consider discussing the aspect of AI semiautomated assistance in designing alveolar scaffolds in cases of resorbed alveol. As todays hydroxyapatite scaffolds can be used for regenerative approach in such deffect, however their manual 3D designing can be complicated. With AI support this step can became effortless. Reference paper https://doi.org/10.3390/ijms232314870

Response 16: Dear reviewer, Thank you for your valuable and effective comments. The related paragraph has been changed as:

 In addition, many methods are being developed for scaffolds used in periodontal regeneration [49]. 3D-printed scaffolds are also a preferred application in the regeneration of alveolar bone defects. Bioceramic-based materials such as hydroxyapatite (HA), β-tricalcium phosphate (β-TCP), and bioactive glass (BG) are widely used to regenerate alveolar bone in the periodontium  [50]. These scaffolds generally provide high mechanical stability and biodegradability, suitable for periodontal regeneration, however, their manual 3D designing can be complicated [51]. The new AI models study can be developed in 3D radiographic data to provide a scaffold for periodontal regeneration in the future.

Point 17: Your Conclusions are quite vague and shall not be after such a long discussion. To improve the conclusions section, consider the following:

Point 18: Summarise the main findings: Start by summarising the main findings of your study in conclusions. In this case, you shall mention that the study focused on using AI to detect periodontal bone loss and found that regional detection was more successful than general detection.

Point 19: Discuss the implications: Explain how the results of your study can contribute to the field of dentistry in general. For example, you could mention that AI can be used as a clinical decision support mechanism to help dentists make more accurate diagnoses.

Point 20: Acknowledge possibly limitations of your study and suggest future research directions. For example, you could mention that the study only examined the assessability of periodontal bone loss by region and that future studies could explore other applications of AI in dentistry.

Point 21: Summarise the main points of your study and reiterate its importance. You can end by highlighting the potential of AI to improve clinical decision making in dentistry and healthcare.

Response 17-18-19-20-21: Dear referee, thank you for your contributions to the conclusion section. We rearranged the discussion section in the article in line with your suggestions. Below we indicate the concluding paragraph.

In our study, it was concluded that regional detection was more successful than total detection in panoramic radiography of periodontal bone loss. The results of the present study have shown that artificial intelligence systems can automatically detect bone destruction. In the future, the use of artificial intelligence in panoramic radiography may provide great convenience in detecting the stage, degree, severity and localized/generalized or molar/incisor distribution of periodontal diseases. It is thought that with the use of this method in routine panoramic radiography, dentists will be able to detect more detailed and easy regional periodontal bone loss in the future. Although the data in our study is limited, new study models can be created that can improve the planning of regenerative approaches in dentistry, especially in the classification and treatment of periodontal disease, by increasing data entry in the future.

  • Language correction support was received from Ozgur Bilgic, whose native language is English

Reviewer 3 Report

In the manuscript entitled “The Yolov5 Approaches to Evaluation of Periodontal Bone Lose: An Artificial Intelligence Study” the authors aimed to evaluate the success of artificial intelligence models in detecting alveolar bone loss by different alveolar regions. The study is interesting. Here some suggestions to improve the quality of the manuscript:

INTRODUCTION

_ Lines 53-54. It is not accurate to say that periodontitis can CAUSE other systemic diseases. Rather I suggest to indicate that periodontal disease is related to several systemic diseases. I suggest mentioning the information contained in these articles (at least):

-          -Cardiovascular diseases: https://doi.org/10.3390/molecules26061777

-          -Diabetes: https://doi.org/10.1155/2022/4955277

-         - Systemic sclerosis: https://doi.org/10.3390/nu13020705

-         - Rheumatoid arthritis: https://doi.org/10.1155/2015/259074

_ Lines 89 and 104: these lines are empty and not needed.

_ I suggest the authors to mention some of the potential uses of CNNs also in 3D imaging including fully automatic segmentation methods. These articles may help to briefly discuss:

-         1 Fully automatic segmentation of the mandible based on convolutional neural networks (CNNs). DOI: 10.1111/ocr.12536

-        - Fully automatic segmentation of sinonasal cavity and pharyngeal airway based on convolutional neural networks. DOI: 10.1016/j.ajodo.2020.05.017

MATERIALS AND METHODS

_ Subparagraphs should be numbered (e.g. 2.1, 2.2, 2.3...) and written in italics. Therefore, remove the bold.

RESULTS

_ Figures 1-10: these figures are too small. I recommend extending them to the full width of the page, as allowed by the template provided by MDPI journals.

DISCUSSION

_ Start this paragraph by mentioning the aim of the study and the main results obtained (1-2 sentences).

_ Lines 285-294: The message as worded is not very clear. The authors should better clarify what are the advantages and disadvantages of periapical radiographs and orthopantomography in the evaluation of the conditions of the supporting alveolar bone.

_ In general, this section is not flowing enough. The authors reported and listed the main findings of several studies (particularly on page 10). The presentation of this section needs to be improved by arguing the topic more critically. Are the results obtained in line with the literature? Are there conflicting results? Are there any potential changes that could affect the diagnostic process in the future?

_ Some final considerations should be made on: limits and potential of the study and future research directions.

Author Response

Dear Reviewer,

We are grateful for the opportunity to revise our manuscript and would like to express our appreciation for the thorough review and insightful feedback provided. We have carefully considered your evaluations and comments, and have made the necessary modifications to enhance the quality and clarity of the manuscript.

We believe that the suggested edits have significantly improved the overall presentation and coherence of the paper. In line with your suggestions, our answers just below your comments were in red; sentences and corrections added to the manuscript were highlighted in the revised text and this response letter.

Thank you again for your time and effort in reviewing our manuscript.

Sincerely,

Dr.UZUN SAYLAN

RESPONSE TO REVIEWER COMMENTS

In the manuscript entitled “The Yolov5 Approaches to Evaluation of Periodontal Bone Lose: An Artificial Intelligence Study” the authors aimed to evaluate the success of artificial intelligence models in detecting alveolar bone loss by different alveolar regions. The study is interesting. Here some suggestions to improve the quality of the manuscript:

INTRODUCTION

Point 1: Lines 53-54. It is not accurate to say that periodontitis can CAUSE other systemic diseases. Rather I suggest to indicate that periodontal disease is related to several systemic diseases. I suggest mentioning the information contained in these articles (at least):

  • -Cardiovascular diseases: https://doi.org/10.3390/molecules26061777
  • -Diabetes: https://doi.org/10.1155/2022/4955277
  • - Systemic sclerosis: https://doi.org/10.3390/nu13020705
  • - Rheumatoid arthritis: https://doi.org/10.1155/2015/259074

Response 1:

For this purpose, stating that periodontal disease is associated with several systemic diseases, the relevant paragraph has been reconstructed with the support of the literature.

It can cause irreversible bone resorption, tooth mobility and tooth loss if not adaquately treated. At the same time, periodontal diseases have the potential to predispose individuals to various systemic diseases such as cardiovascular disease, oral and colorectal cancer, gastrointestinal diseases, respiratory tract infection and pneumonia, adverse pregnancy outcomes, diabetes and insulin resistance, and Alzheimer's disease (Bui FQ, 2019).

Point 2: Lines 89 and 104: these lines are empty and not needed.

Response 2: Related empty lines were removed.

Point 3: I suggest the authors to mention some of the potential uses of CNNs also in 3D imaging including fully automatic segmentation methods. These articles may help to briefly discuss:

  • 1 Fully automatic segmentation of the mandible based on convolutional neural networks (CNNs). DOI: 10.1111/ocr.12536
  • - Fully automatic segmentation of sinonasal cavity and pharyngeal airway based on convolutional neural networks. DOI: 10.1016/j.ajodo.2020.05.017

Response 3: Thanks to your valuable contributions, the section below the introduction has been added.

Although three-dimensional (3D) evaluations are widely used in various fields of dentistry, including implantology, surgery, endodontics, and orthodontics, their application in periodontology is primarily limited to the assessment of furcations, craters, and bone defects, as well as the determination of root form and alveolar relationship (Harvey S, 2020). During a standard periodontal evaluation, periapical, bite-wing, and panoramic radiography are the preferred methods for assessing the level of alveolar bone in the interproximal area. This is due to the cost-effectiveness, rapidity, and lower radiation exposure associated with these 2D imaging techniques, when compared to 3D imaging modalities (Eshraangi TL, 2012). In consideration of these factors, we opted to employ panoramic radiography images, which are commonly utilized in periodontal assessments for evaluating alveolar bone loss, in our study.

MATERIALS AND METHODS

Point 4: Subparagraphs should be numbered (e.g. 2.1, 2.2, 2.3...) and written in italics. Therefore, remove the bold.

Response 4: Subparagraphs were numbered and written in italics, and the bold are removed as per your suggestion.

RESULTS

Point 5: Figures 1-10: these figures are too small. I recommend extending them to the full width of the page, as allowed by the template provided by MDPI journals.

Response 5: Figures 1-10 were extended to the full width of the page, as allowed by the template provided by MDPI journals, as per your suggestions.

DISCUSSION

Point 6: Start this paragraph by mentioning the aim of the study and the main results obtained (1-2 sentences).

Response 6: Dear reviewer, thank you for your suggestion. At your request, we have chosen to start the discussion section with a paragraph that talks about the main purpose and results of the study:

 The study aimed to evaluate the success of artificial intelligence models in detecting alveolar bone loss by different alveolar regions. It was determined that local alveolar bone loss detection was more successful than general alveolar bone loss detection in all regions.

Point 7: Lines 285-294: The message as worded is not very clear. The authors should better clarify what are the advantages and disadvantages of periapical radiographs and orthopantomography in the evaluation of the conditions of the supporting alveolar bone.

Response 7:  Dear reviewer, the relevant paragraph has been rephrased to more clearly demonstrate the advantages and limitations of using 2D techniques.

Two-dimensional radiographic examination by means of peri-apical radiography, is still the standard method for assessing marginal bone loss. In addition, decay, root morphology and resorptions can be dxidentifed (Papapanou PN, 2000). Panoramic radiographs may occasionally be combined with peri-apical radiographs as an alternative to a fullmouth series of peri-apical radiographs to reduce the total radiation dose(Berghuis G, 2021).

Point 8: In general, this section is not flowing enough. The authors reported and listed the main findings of several studies (particularly on page 10). The presentation of this section needs to be improved by arguing the topic more critically. Are the results obtained in line with the literature? Are there conflicting results? Are there any potential changes that could affect the diagnostic process in the future?

Response 8: In line with your suggestion, the relevant part of the article was rephrased in terms of similarities, differences and limitations.

Point 9: Some final considerations should be made on: limits and potential of the study and future research directions.

Response 9: We have edited the conclusion section in line with your suggestions.

In our study, it was concluded that regional detection was more successful than total detection in panoramic radiography of periodontal bone loss. The results of the present study have shown that artificial intelligence systems can automatically detect bone destruction. In the future, the use of artificial intelligence in panoramic radiography may provide great convenience in detecting the stage, degree, severity and localized/generalized or molar/incisor distribution of periodontal diseases. It is thought that with the use of this method in routine panoramic radiography, dentists will be able to detect more detailed and easy regional periodontal bone loss in the future. Although the data in our study is limited, new study models can be created that can improve the planning of regenerative approaches in dentistry, especially in the classification and treatment of periodontal disease, by increasing data entry in the future.

  • Language correction support was received from Ozgur Bilgic, whose native language is English.

Round 2

Reviewer 1 Report

diagnostics-2193263R1

The Yolov5 approaches to evaluation of periodontal bone lose: an artificial intelligence study

In 685 panoramic radiographs artificial intelligence (AI) was used to assess whether periodontal bone loss (PBL: distance cementoenamel junction to alveolar crest > 2 mm) was present or not. Sensitivity and F1 score should be assessed. However, no gold standard was defined to assess sensitivity of the AI method.

This is the revision of a manuscript on a diagnostic study on the highly relevant issue of AI approaches to simplify diagnostic processes in periodontology. The authors have improved the manuscript. However, there are major issues and questions that  have to be clarified.

Comments:

In general

The authors have set the threshold for PBL, as distance between cemento-enamel junction (CEJ) to alveolar crest (AC) > 2 mm. However, the 2018 Classification of Periodontal and Peri-implant Diseases and Conditions sets the threshold at 3 mm (Chapple et al. 2018).

The authors have to explain the different meaning of sensitivity in the context of AI at large to prevent misunderstanding.

In dentistry directions as “below” (mandible: below the CEJ) or “above” (maxilla: above the CEJ) do not make any sense related to teeth. This is the reason why dentistry uses terms as “mesial”, “distal”, “apical” or “coronal” to give directions related to teeth. The authors may wish to use the respective professional terms: i.e., AC apical of CEJ instead of “below” or “above”.

Abstract

Page 1, lines 21-22: The authors may wish to explicitly express that they are aiming to assess PBL yes or no.

Introduction

Page 3, lines 116-118: Add “(yes/no”) after “alveolar bone loss”.

Material & Methods

Pages 4, lines 156-158: This sentence does not make sense. The authors set the threshold for PBL at a distance > 2 mm between CEJ and AC. This means that if AI detects a distance > 2 mm between CEJ/AC it will signal PBL. However, the 2018 Classification of Periodontal and Peri-implant Diseases and Conditions sets the threshold at 3 mm (Chapple et al. 2018). This classification has been published more than 4 years ago and should be considered for the detection of PBL.

Discussion

Page 12, lines 288-290: This study simply assesses whether PBL is present or not. An extrapolation to AI guided construction of scaffolds for regeneration of periodontal defects is far beyond the published data. Delete paragraph.

Page 12, lines 411-414: This study simply assesses whether PBL is present or not. The authors repeatedly give the response that they did not assess severity of periodontal disease. However, now they conclude that AI “can provide a great convenience in detecting the stage, degree, severity, and localized/generalized or molar/incisor distribution of periodontal diseases”? This conclusion is also far beyond the published data. Delete sentence.

Data availability

Page 13, lines 434: If the authors do not want to share open access data they may state this clearly. The term “not applicable” is not appropriate.diagnostics-2193263R1

The Yolov5 approaches to evaluation of periodontal bone lose: an artificial intelligence study

In 685 panoramic radiographs artificial intelligence (AI) was used to assess whether periodontal bone loss (PBL: distance cementoenamel junction to alveolar crest > 2 mm) was present or not. Sensitivity and F1 score should be assessed. However, no gold standard was defined to assess sensitivity of the AI method.

This is the revision of a manuscript on a diagnostic study on the highly relevant issue of AI approaches to simplify diagnostic processes in periodontology. The authors have improved the manuscript. However, there are major issues and questions that  have to be clarified.

Comments:

In general

The authors have set the threshold for PBL, as distance between cemento-enamel junction (CEJ) to alveolar crest (AC) > 2 mm. However, the 2018 Classification of Periodontal and Peri-implant Diseases and Conditions sets the threshold at 3 mm (Chapple et al. 2018).

The authors have to explain the different meaning of sensitivity in the context of AI at large to prevent misunderstanding.

In dentistry directions as “below” (mandible: below the CEJ) or “above” (maxilla: above the CEJ) do not make any sense related to teeth. This is the reason why dentistry uses terms as “mesial”, “distal”, “apical” or “coronal” to give directions related to teeth. The authors may wish to use the respective professional terms: i.e., AC apical of CEJ instead of “below” or “above”.

Abstract

Page 1, lines 21-22: The authors may wish to explicitly express that they are aiming to assess PBL yes or no.

Introduction

Page 3, lines 116-118: Add “(yes/no”) after “alveolar bone loss”.

Material & Methods

Pages 4, lines 156-158: This sentence does not make sense. The authors set the threshold for PBL at a distance > 2 mm between CEJ and AC. This means that if AI detects a distance > 2 mm between CEJ/AC it will signal PBL. However, the 2018 Classification of Periodontal and Peri-implant Diseases and Conditions sets the threshold at 3 mm (Chapple et al. 2018). This classification has been published more than 4 years ago and should be considered for the detection of PBL.

Discussion

Page 12, lines 288-290: This study simply assesses whether PBL is present or not. An extrapolation to AI guided construction of scaffolds for regeneration of periodontal defects is far beyond the published data. Delete paragraph.

Page 12, lines 411-414: This study simply assesses whether PBL is present or not. The authors repeatedly give the response that they did not assess severity of periodontal disease. However, now they conclude that AI “can provide a great convenience in detecting the stage, degree, severity, and localized/generalized or molar/incisor distribution of periodontal diseases”? This conclusion is also far beyond the published data. Delete sentence.

Data availability

Page 13, lines 434: If the authors do not want to share open access data they may state this clearly. The term “not applicable” is not appropriate.diagnostics-2193263R1

The Yolov5 approaches to evaluation of periodontal bone lose: an artificial intelligence study

In 685 panoramic radiographs artificial intelligence (AI) was used to assess whether periodontal bone loss (PBL: distance cementoenamel junction to alveolar crest > 2 mm) was present or not. Sensitivity and F1 score should be assessed. However, no gold standard was defined to assess sensitivity of the AI method.

This is the revision of a manuscript on a diagnostic study on the highly relevant issue of AI approaches to simplify diagnostic processes in periodontology. The authors have improved the manuscript. However, there are major issues and questions that  have to be clarified.

Comments:

In general

The authors have set the threshold for PBL, as distance between cemento-enamel junction (CEJ) to alveolar crest (AC) > 2 mm. However, the 2018 Classification of Periodontal and Peri-implant Diseases and Conditions sets the threshold at 3 mm (Chapple et al. 2018).

The authors have to explain the different meaning of sensitivity in the context of AI at large to prevent misunderstanding.

In dentistry directions as “below” (mandible: below the CEJ) or “above” (maxilla: above the CEJ) do not make any sense related to teeth. This is the reason why dentistry uses terms as “mesial”, “distal”, “apical” or “coronal” to give directions related to teeth. The authors may wish to use the respective professional terms: i.e., AC apical of CEJ instead of “below” or “above”.

Abstract

Page 1, lines 21-22: The authors may wish to explicitly express that they are aiming to assess PBL yes or no.

Introduction

Page 3, lines 116-118: Add “(yes/no”) after “alveolar bone loss”.

Material & Methods

Pages 4, lines 156-158: This sentence does not make sense. The authors set the threshold for PBL at a distance > 2 mm between CEJ and AC. This means that if AI detects a distance > 2 mm between CEJ/AC it will signal PBL. However, the 2018 Classification of Periodontal and Peri-implant Diseases and Conditions sets the threshold at 3 mm (Chapple et al. 2018). This classification has been published more than 4 years ago and should be considered for the detection of PBL.

Discussion

Page 12, lines 288-290: This study simply assesses whether PBL is present or not. An extrapolation to AI guided construction of scaffolds for regeneration of periodontal defects is far beyond the published data. Delete paragraph.

Page 12, lines 411-414: This study simply assesses whether PBL is present or not. The authors repeatedly give the response that they did not assess severity of periodontal disease. However, now they conclude that AI “can provide a great convenience in detecting the stage, degree, severity, and localized/generalized or molar/incisor distribution of periodontal diseases”? This conclusion is also far beyond the published data. Delete sentence.

Data availability

Page 13, lines 434: If the authors do not want to share open access data they may state this clearly. The term “not applicable” is not appropriate.diagnostics-2193263R1

The Yolov5 approaches to evaluation of periodontal bone lose: an artificial intelligence study

In 685 panoramic radiographs artificial intelligence (AI) was used to assess whether periodontal bone loss (PBL: distance cementoenamel junction to alveolar crest > 2 mm) was present or not. Sensitivity and F1 score should be assessed. However, no gold standard was defined to assess sensitivity of the AI method.

This is the revision of a manuscript on a diagnostic study on the highly relevant issue of AI approaches to simplify diagnostic processes in periodontology. The authors have improved the manuscript. However, there are major issues and questions that  have to be clarified.

Comments:

In general

The authors have set the threshold for PBL, as distance between cemento-enamel junction (CEJ) to alveolar crest (AC) > 2 mm. However, the 2018 Classification of Periodontal and Peri-implant Diseases and Conditions sets the threshold at 3 mm (Chapple et al. 2018).

The authors have to explain the different meaning of sensitivity in the context of AI at large to prevent misunderstanding.

In dentistry directions as “below” (mandible: below the CEJ) or “above” (maxilla: above the CEJ) do not make any sense related to teeth. This is the reason why dentistry uses terms as “mesial”, “distal”, “apical” or “coronal” to give directions related to teeth. The authors may wish to use the respective professional terms: i.e., AC apical of CEJ instead of “below” or “above”.

Abstract

Page 1, lines 21-22: The authors may wish to explicitly express that they are aiming to assess PBL yes or no.

Introduction

Page 3, lines 116-118: Add “(yes/no”) after “alveolar bone loss”.

Material & Methods

Pages 4, lines 156-158: This sentence does not make sense. The authors set the threshold for PBL at a distance > 2 mm between CEJ and AC. This means that if AI detects a distance > 2 mm between CEJ/AC it will signal PBL. However, the 2018 Classification of Periodontal and Peri-implant Diseases and Conditions sets the threshold at 3 mm (Chapple et al. 2018). This classification has been published more than 4 years ago and should be considered for the detection of PBL.

Discussion

Page 12, lines 288-290: This study simply assesses whether PBL is present or not. An extrapolation to AI guided construction of scaffolds for regeneration of periodontal defects is far beyond the published data. Delete paragraph.

Page 12, lines 411-414: This study simply assesses whether PBL is present or not. The authors repeatedly give the response that they did not assess severity of periodontal disease. However, now they conclude that AI “can provide a great convenience in detecting the stage, degree, severity, and localized/generalized or molar/incisor distribution of periodontal diseases”? This conclusion is also far beyond the published data. Delete sentence.

Data availability

Page 13, lines 434: If the authors do not want to share open access data they may state this clearly. The term “not applicable” is not appropriate.diagnostics-2193263R1

The Yolov5 approaches to evaluation of periodontal bone lose: an artificial intelligence study

In 685 panoramic radiographs artificial intelligence (AI) was used to assess whether periodontal bone loss (PBL: distance cementoenamel junction to alveolar crest > 2 mm) was present or not. Sensitivity and F1 score should be assessed. However, no gold standard was defined to assess sensitivity of the AI method.

This is the revision of a manuscript on a diagnostic study on the highly relevant issue of AI approaches to simplify diagnostic processes in periodontology. The authors have improved the manuscript. However, there are major issues and questions that  have to be clarified.

Comments:

In general

The authors have set the threshold for PBL, as distance between cemento-enamel junction (CEJ) to alveolar crest (AC) > 2 mm. However, the 2018 Classification of Periodontal and Peri-implant Diseases and Conditions sets the threshold at 3 mm (Chapple et al. 2018).

The authors have to explain the different meaning of sensitivity in the context of AI at large to prevent misunderstanding.

In dentistry directions as “below” (mandible: below the CEJ) or “above” (maxilla: above the CEJ) do not make any sense related to teeth. This is the reason why dentistry uses terms as “mesial”, “distal”, “apical” or “coronal” to give directions related to teeth. The authors may wish to use the respective professional terms: i.e., AC apical of CEJ instead of “below” or “above”.

Abstract

Page 1, lines 21-22: The authors may wish to explicitly express that they are aiming to assess PBL yes or no.

Introduction

Page 3, lines 116-118: Add “(yes/no”) after “alveolar bone loss”.

Material & Methods

Pages 4, lines 156-158: This sentence does not make sense. The authors set the threshold for PBL at a distance > 2 mm between CEJ and AC. This means that if AI detects a distance > 2 mm between CEJ/AC it will signal PBL. However, the 2018 Classification of Periodontal and Peri-implant Diseases and Conditions sets the threshold at 3 mm (Chapple et al. 2018). This classification has been published more than 4 years ago and should be considered for the detection of PBL.

Discussion

Page 12, lines 288-290: This study simply assesses whether PBL is present or not. An extrapolation to AI guided construction of scaffolds for regeneration of periodontal defects is far beyond the published data. Delete paragraph.

Page 12, lines 411-414: This study simply assesses whether PBL is present or not. The authors repeatedly give the response that they did not assess severity of periodontal disease. However, now they conclude that AI “can provide a great convenience in detecting the stage, degree, severity, and localized/generalized or molar/incisor distribution of periodontal diseases”? This conclusion is also far beyond the published data. Delete sentence.

Data availability

Page 13, lines 434: If the authors do not want to share open access data they may state this clearly. The term “not applicable” is not appropriate.diagnostics-2193263R1

The Yolov5 approaches to evaluation of periodontal bone lose: an artificial intelligence study

In 685 panoramic radiographs artificial intelligence (AI) was used to assess whether periodontal bone loss (PBL: distance cementoenamel junction to alveolar crest > 2 mm) was present or not. Sensitivity and F1 score should be assessed. However, no gold standard was defined to assess sensitivity of the AI method.

This is the revision of a manuscript on a diagnostic study on the highly relevant issue of AI approaches to simplify diagnostic processes in periodontology. The authors have improved the manuscript. However, there are major issues and questions that  have to be clarified.

Comments:

In general

The authors have set the threshold for PBL, as distance between cemento-enamel junction (CEJ) to alveolar crest (AC) > 2 mm. However, the 2018 Classification of Periodontal and Peri-implant Diseases and Conditions sets the threshold at 3 mm (Chapple et al. 2018).

The authors have to explain the different meaning of sensitivity in the context of AI at large to prevent misunderstanding.

In dentistry directions as “below” (mandible: below the CEJ) or “above” (maxilla: above the CEJ) do not make any sense related to teeth. This is the reason why dentistry uses terms as “mesial”, “distal”, “apical” or “coronal” to give directions related to teeth. The authors may wish to use the respective professional terms: i.e., AC apical of CEJ instead of “below” or “above”.

Abstract

Page 1, lines 21-22: The authors may wish to explicitly express that they are aiming to assess PBL yes or no.

Introduction

Page 3, lines 116-118: Add “(yes/no”) after “alveolar bone loss”.

Material & Methods

Pages 4, lines 156-158: This sentence does not make sense. The authors set the threshold for PBL at a distance > 2 mm between CEJ and AC. This means that if AI detects a distance > 2 mm between CEJ/AC it will signal PBL. However, the 2018 Classification of Periodontal and Peri-implant Diseases and Conditions sets the threshold at 3 mm (Chapple et al. 2018). This classification has been published more than 4 years ago and should be considered for the detection of PBL.

Discussion

Page 12, lines 288-290: This study simply assesses whether PBL is present or not. An extrapolation to AI guided construction of scaffolds for regeneration of periodontal defects is far beyond the published data. Delete paragraph.

Page 12, lines 411-414: This study simply assesses whether PBL is present or not. The authors repeatedly give the response that they did not assess severity of periodontal disease. However, now they conclude that AI “can provide a great convenience in detecting the stage, degree, severity, and localized/generalized or molar/incisor distribution of periodontal diseases”? This conclusion is also far beyond the published data. Delete sentence.

Data availability

Page 13, lines 434: If the authors do not want to share open access data they may state this clearly. The term “not applicable” is not appropriate.diagnostics-2193263R1

The Yolov5 approaches to evaluation of periodontal bone lose: an artificial intelligence study

In 685 panoramic radiographs artificial intelligence (AI) was used to assess whether periodontal bone loss (PBL: distance cementoenamel junction to alveolar crest > 2 mm) was present or not. Sensitivity and F1 score should be assessed. However, no gold standard was defined to assess sensitivity of the AI method.

This is the revision of a manuscript on a diagnostic study on the highly relevant issue of AI approaches to simplify diagnostic processes in periodontology. The authors have improved the manuscript. However, there are major issues and questions that  have to be clarified.

Comments:

In general

The authors have set the threshold for PBL, as distance between cemento-enamel junction (CEJ) to alveolar crest (AC) > 2 mm. However, the 2018 Classification of Periodontal and Peri-implant Diseases and Conditions sets the threshold at 3 mm (Chapple et al. 2018).

The authors have to explain the different meaning of sensitivity in the context of AI at large to prevent misunderstanding.

In dentistry directions as “below” (mandible: below the CEJ) or “above” (maxilla: above the CEJ) do not make any sense related to teeth. This is the reason why dentistry uses terms as “mesial”, “distal”, “apical” or “coronal” to give directions related to teeth. The authors may wish to use the respective professional terms: i.e., AC apical of CEJ instead of “below” or “above”.

Abstract

Page 1, lines 21-22: The authors may wish to explicitly express that they are aiming to assess PBL yes or no.

Introduction

Page 3, lines 116-118: Add “(yes/no”) after “alveolar bone loss”.

Material & Methods

Pages 4, lines 156-158: This sentence does not make sense. The authors set the threshold for PBL at a distance > 2 mm between CEJ and AC. This means that if AI detects a distance > 2 mm between CEJ/AC it will signal PBL. However, the 2018 Classification of Periodontal and Peri-implant Diseases and Conditions sets the threshold at 3 mm (Chapple et al. 2018). This classification has been published more than 4 years ago and should be considered for the detection of PBL.

Discussion

Page 12, lines 288-290: This study simply assesses whether PBL is present or not. An extrapolation to AI guided construction of scaffolds for regeneration of periodontal defects is far beyond the published data. Delete paragraph.

Page 12, lines 411-414: This study simply assesses whether PBL is present or not. The authors repeatedly give the response that they did not assess severity of periodontal disease. However, now they conclude that AI “can provide a great convenience in detecting the stage, degree, severity, and localized/generalized or molar/incisor distribution of periodontal diseases”? This conclusion is also far beyond the published data. Delete sentence.

Data availability

Page 13, lines 434: If the authors do not want to share open access data they may state this clearly. The term “not applicable” is not appropriate.diagnostics-2193263R1

The Yolov5 approaches to evaluation of periodontal bone lose: an artificial intelligence study

In 685 panoramic radiographs artificial intelligence (AI) was used to assess whether periodontal bone loss (PBL: distance cementoenamel junction to alveolar crest > 2 mm) was present or not. Sensitivity and F1 score should be assessed. However, no gold standard was defined to assess sensitivity of the AI method.

This is the revision of a manuscript on a diagnostic study on the highly relevant issue of AI approaches to simplify diagnostic processes in periodontology. The authors have improved the manuscript. However, there are major issues and questions that  have to be clarified.

Comments:

In general

The authors have set the threshold for PBL, as distance between cemento-enamel junction (CEJ) to alveolar crest (AC) > 2 mm. However, the 2018 Classification of Periodontal and Peri-implant Diseases and Conditions sets the threshold at 3 mm (Chapple et al. 2018).

The authors have to explain the different meaning of sensitivity in the context of AI at large to prevent misunderstanding.

In dentistry directions as “below” (mandible: below the CEJ) or “above” (maxilla: above the CEJ) do not make any sense related to teeth. This is the reason why dentistry uses terms as “mesial”, “distal”, “apical” or “coronal” to give directions related to teeth. The authors may wish to use the respective professional terms: i.e., AC apical of CEJ instead of “below” or “above”.

Abstract

Page 1, lines 21-22: The authors may wish to explicitly express that they are aiming to assess PBL yes or no.

Introduction

Page 3, lines 116-118: Add “(yes/no”) after “alveolar bone loss”.

Material & Methods

Pages 4, lines 156-158: This sentence does not make sense. The authors set the threshold for PBL at a distance > 2 mm between CEJ and AC. This means that if AI detects a distance > 2 mm between CEJ/AC it will signal PBL. However, the 2018 Classification of Periodontal and Peri-implant Diseases and Conditions sets the threshold at 3 mm (Chapple et al. 2018). This classification has been published more than 4 years ago and should be considered for the detection of PBL.

Discussion

Page 12, lines 288-290: This study simply assesses whether PBL is present or not. An extrapolation to AI guided construction of scaffolds for regeneration of periodontal defects is far beyond the published data. Delete paragraph.

Page 12, lines 411-414: This study simply assesses whether PBL is present or not. The authors repeatedly give the response that they did not assess severity of periodontal disease. However, now they conclude that AI “can provide a great convenience in detecting the stage, degree, severity, and localized/generalized or molar/incisor distribution of periodontal diseases”? This conclusion is also far beyond the published data. Delete sentence.

Data availability

Page 13, lines 434: If the authors do not want to share open access data they may state this clearly. The term “not applicable” is not appropriate.diagnostics-2193263R1

The Yolov5 approaches to evaluation of periodontal bone lose: an artificial intelligence study

In 685 panoramic radiographs artificial intelligence (AI) was used to assess whether periodontal bone loss (PBL: distance cementoenamel junction to alveolar crest > 2 mm) was present or not. Sensitivity and F1 score should be assessed. However, no gold standard was defined to assess sensitivity of the AI method.

This is the revision of a manuscript on a diagnostic study on the highly relevant issue of AI approaches to simplify diagnostic processes in periodontology. The authors have improved the manuscript. However, there are major issues and questions that  have to be clarified.

Comments:

In general

The authors have set the threshold for PBL, as distance between cemento-enamel junction (CEJ) to alveolar crest (AC) > 2 mm. However, the 2018 Classification of Periodontal and Peri-implant Diseases and Conditions sets the threshold at 3 mm (Chapple et al. 2018).

The authors have to explain the different meaning of sensitivity in the context of AI at large to prevent misunderstanding.

In dentistry directions as “below” (mandible: below the CEJ) or “above” (maxilla: above the CEJ) do not make any sense related to teeth. This is the reason why dentistry uses terms as “mesial”, “distal”, “apical” or “coronal” to give directions related to teeth. The authors may wish to use the respective professional terms: i.e., AC apical of CEJ instead of “below” or “above”.

Abstract

Page 1, lines 21-22: The authors may wish to explicitly express that they are aiming to assess PBL yes or no.

Introduction

Page 3, lines 116-118: Add “(yes/no”) after “alveolar bone loss”.

Material & Methods

Pages 4, lines 156-158: This sentence does not make sense. The authors set the threshold for PBL at a distance > 2 mm between CEJ and AC. This means that if AI detects a distance > 2 mm between CEJ/AC it will signal PBL. However, the 2018 Classification of Periodontal and Peri-implant Diseases and Conditions sets the threshold at 3 mm (Chapple et al. 2018). This classification has been published more than 4 years ago and should be considered for the detection of PBL.

Discussion

Page 12, lines 288-290: This study simply assesses whether PBL is present or not. An extrapolation to AI guided construction of scaffolds for regeneration of periodontal defects is far beyond the published data. Delete paragraph.

Page 12, lines 411-414: This study simply assesses whether PBL is present or not. The authors repeatedly give the response that they did not assess severity of periodontal disease. However, now they conclude that AI “can provide a great convenience in detecting the stage, degree, severity, and localized/generalized or molar/incisor distribution of periodontal diseases”? This conclusion is also far beyond the published data. Delete sentence.

Data availability

Page 13, lines 434: If the authors do not want to share open access data they may state this clearly. The term “not applicable” is not appropriate.diagnostics-2193263R1

The Yolov5 approaches to evaluation of periodontal bone lose: an artificial intelligence study

In 685 panoramic radiographs artificial intelligence (AI) was used to assess whether periodontal bone loss (PBL: distance cementoenamel junction to alveolar crest > 2 mm) was present or not. Sensitivity and F1 score should be assessed. However, no gold standard was defined to assess sensitivity of the AI method.

This is the revision of a manuscript on a diagnostic study on the highly relevant issue of AI approaches to simplify diagnostic processes in periodontology. The authors have improved the manuscript. However, there are major issues and questions that  have to be clarified.

Comments:

In general

The authors have set the threshold for PBL, as distance between cemento-enamel junction (CEJ) to alveolar crest (AC) > 2 mm. However, the 2018 Classification of Periodontal and Peri-implant Diseases and Conditions sets the threshold at 3 mm (Chapple et al. 2018).

The authors have to explain the different meaning of sensitivity in the context of AI at large to prevent misunderstanding.

In dentistry directions as “below” (mandible: below the CEJ) or “above” (maxilla: above the CEJ) do not make any sense related to teeth. This is the reason why dentistry uses terms as “mesial”, “distal”, “apical” or “coronal” to give directions related to teeth. The authors may wish to use the respective professional terms: i.e., AC apical of CEJ instead of “below” or “above”.

Abstract

Page 1, lines 21-22: The authors may wish to explicitly express that they are aiming to assess PBL yes or no.

Introduction

Page 3, lines 116-118: Add “(yes/no”) after “alveolar bone loss”.

Material & Methods

Pages 4, lines 156-158: This sentence does not make sense. The authors set the threshold for PBL at a distance > 2 mm between CEJ and AC. This means that if AI detects a distance > 2 mm between CEJ/AC it will signal PBL. However, the 2018 Classification of Periodontal and Peri-implant Diseases and Conditions sets the threshold at 3 mm (Chapple et al. 2018). This classification has been published more than 4 years ago and should be considered for the detection of PBL.

Discussion

Page 12, lines 288-290: This study simply assesses whether PBL is present or not. An extrapolation to AI guided construction of scaffolds for regeneration of periodontal defects is far beyond the published data. Delete paragraph.

Page 12, lines 411-414: This study simply assesses whether PBL is present or not. The authors repeatedly give the response that they did not assess severity of periodontal disease. However, now they conclude that AI “can provide a great convenience in detecting the stage, degree, severity, and localized/generalized or molar/incisor distribution of periodontal diseases”? This conclusion is also far beyond the published data. Delete sentence.

Data availability

Page 13, lines 434: If the authors do not want to share open access data they may state this clearly. The term “not applicable” is not appropriate.

Author Response

Response to Reviewer Comments (Revision-2)

REPORT

In 685 panoramic radiographs artificial intelligence (AI) was used to assess whether periodontal bone loss (PBL: distance cementoenamel junction to alveolar crest > 2 mm) was present or not. Sensitivity and F1 score should be assessed. However, no gold standard was defined to assess sensitivity of the AI method.

This is the revision of a manuscript on a diagnostic study on the highly relevant issue of AI approaches to simplify diagnostic processes in periodontology. The authors have improved the manuscript. However, there are major issues and questions that have to be clarified.

Response: Dear referee, thank you for your evaluation, I hope we understood your question correctly. We tried to explain in line with your suggestion and added the sentence in quotation marks to the article.

In segmentation studies, sensitivity is the metric that evaluates the success of the model. To determine the sensitivity of artificial intelligence models conducted on radiographs, a gold standard is an acceptable reference standard used to determine the accuracy of a test.

The gold standard for radiographs is typically based on the visual interpretation of the radiologist who independently reviewed the radiographs. There are many artificial intelligence studies using similar methods for evaluating of model performance in the literature. This is considered the gold standard by which the results of the radiologist can be compared with the results of the AI model and its sensitivity can be evaluated.

“In this study, the test data used to evaluate the success of the artificial intelligence model were evaluated by the radiologist (EB) and accepted as the gold standard in determining the sensitivity values.”

Clinical evaluation and Cone Beam Computer Tomography (CBCT) evaluation were not performed in the study, and this was mentioned in the limitations of the study.

Comments:

In general

Point 1: The authors have set the threshold for PBL, as distance between cemento-enamel junction (CEJ) to alveolar crest (AC) > 2 mm. However, the 2018 Classification of Periodontal and Peri-implant Diseases and Conditions sets the threshold at 3 mm (Chapple et al. 2018).

Response 1:  The purpose of this study is not to classify periodontal diseases. In our study, based on a certain reference point, both total and segmental bone destruction is labeled and introduced to artificial intelligence. Thus, the aim is to show the segmental or total bone destruction to the physician through artificial intelligence.

In our study, 2 mm apical of the cemento-enamel junction was determined as the reference point in the radiography.

Following are the articles that used reference point determination:

-Kim SH, Kim J, Yang S, Oh SH, Lee SP, Yang HJ, Kim TI, Yi WJ. Automatic and quantitative measurement of alveolar bone level in OCT images using deep learning. Biomed Opt Express. 2022 Sep 26;13(10):5468-5482. doi: 10.1364/BOE.468212. PMID: 36425614; PMCID: PMC9664875.

Accurate assessment of the ABL is important for diagnosis and progression of periodontitis. Generally, when the ABL is greater than 2 mm, it is considered alveolar bone loss.”

-Helmi MF, Huang H, Goodson JM, Hasturk H, Tavares M, Natto ZS. Prevalence of periodontitis and alveolar bone loss in a patient population at Harvard School of Dental Medicine. BMC Oral Health. 2019 Nov 21;19(1):254. doi: 10.1186/s12903-019-0925-z. PMID: 31752793; PMCID: PMC6873420.

Outcome assessment: Radiographic indication of interproximal bone loss occurs when the distance between the CEJ and the alveolar bone crest is greater than or equal to 2 mm, as determined on a bitewing radiograph”.

-Wylleman A, Van der Veken D, Teughels W, Quirynen M, Laleman I. Alveolar bone level at deciduous molars in Flemish children: A retrospective, radiographic study. J Clin Periodontol. 2020 Jun;47(6):660-667. doi: 10.1111/jcpe.13280. Epub 2020 Mar 19. PMID: 32144794.

Interestingly, some authors use a 2 mm CEJ-ABC distance, while others choose a 3 mm cut-off value to correct for radiographic imprecisions.  In the current study, a  2  mm  cut-off  point  was  used  to  define bone loss.”

-Castro LO, Castro IO, de Alencar AH, Valladares-Neto J, Estrela C. Cone beam computed tomography evaluation of distance from cementoenamel junction to alveolar crest before and after nonextraction orthodontic treatment. Angle Orthod. 2016 Jul;86(4):543-9. doi: 10.2319/040815-235.1. Epub 2015 Sep 17. PMID: 26379114; PMCID: PMC8601496.

“When the distance from the cementoenamel junction to the bone crest was shorter than or equal to 2 mm, no alveolar bone defects were recorded because this distance had to be greater than 2 mm to be classified as alveolar bone dehiscence.”

-de Toledo BE, Barroso EM, Martins AT, Zuza EP. Prevalence of Periodontal Bone Loss in Brazilian Adolescents through Interproximal Radiography. Int J Dent. 2012;2012:357056. doi: 10.1155/2012/357056. Epub 2012 Sep 29. PMID: 23056048; PMCID: PMC3465972.

A distance between the cementoenamel junction (CEJ) and the alveolar bone crest more than 2 mm was considered as periodontal bone loss.”

-Semenoff L, Semenoff TA, Pedro FL, Volpato ER, Machado MA, Borges AH, Semenoff-Segundo A. Are panoramic radiographs reliable to diagnose mild alveolar bone resorption? ISRN Dent. 2011;2011:363578. doi: 10.5402/2011/363578. Epub 2011 May 4. PMID: 21991470; PMCID: PMC3169835.

“The observed measurements for both conventional and digitized radiographs were classified according to the following categories: 0–2 mm (absence of bone loss), 3–5 mm (moderate bone loss), and ≥6 mm (advanced bone loss) and compared among them”

Point 2: The authors have to explain the different meaning of sensitivity in the context of AI at large to prevent misunderstanding.

Dear referee, thank you for your suggestion, the sentence below has been added to the article for your suggested correction.

Response 2: Sensitivity represents the proportion of actual positive cases that the model has correctly identified as positive. Precision is another metric that measures how accurate the results are. It indicates how many of the positively classified examples by the model are truly positive. In other words, in the case of a disease or dental condition, precision shows how accurately the model has classified the disease examples as positive. Precision is calculated only on the positively classified examples. F1 score is the harmonic mean of precision and sensitivity. It takes into account both precision and sensitivity, and is a way to balance the trade-off between them. A high F1 score means that both precision and sensitivity are high, which indicates a model with a good balance between detecting true positives and avoiding false positives.

Point 3: In dentistry directions as “below” (mandible: below the CEJ) or “above” (maxilla: above the CEJ) do not make any sense related to teeth. This is the reason why dentistry uses terms as “mesial”, “distal”, “apical” or “coronal” to give directions related to teeth. The authors may wish to use the respective professional terms: i.e., AC apical of CEJ instead of “below” or “above”.

Response 3: Dear referee, 'below' has been changed to 'apical' as per your suggestion.

Abstract

Point 4: Page 1, lines 21-22: The authors may wish to explicitly express that they are aiming to assess PBL yes or no.

Response 4: Dear reviwer, Thank you for your suggestion. In the abstract, it is clearly stated that the PBL estimation is evaluated by adding `This study aims to evaluate the effectiveness of AI models in identifying alveolar bone loss as present or absent across different regions. To achieve this goal, alveolar bone loss models were generated using the PyTorch-based YOLO-v5 model implemented via CranioCatch software, detecting periodontal bone loss areas and labeling them using segmentation method on 685 panoramic radiographs` to the specified line.

Introduction

Point 5: Page 3, lines 116-118: Add “(yes/no”) after “alveolar bone loss”.

Response 5: In line with your suggestion, the expression 'present or absent' has been added after the expression of alveolar bone loss.

Material & Methods

Point 6: Pages 4, lines 156-158: This sentence does not make sense. The authors set the threshold for PBL at a distance > 2 mm between CEJ and AC. This means that if AI detects a distance > 2 mm between CEJ/AC it will signal PBL. However, the 2018 Classification of Periodontal and Peri-implant Diseases and Conditions sets the threshold at 3 mm (Chapple et al. 2018). This classification has been published more than 4 years ago and should be considered for the detection of PBL.

Response 6. Our main aim in our study was to introduce total and segmental bone destruction to artificial intelligence. Rather than calculating the periodontal classification, the percentage of destruction bone, our priority was to show the physician as present/absent total or segmental bone destruction in the first stage through the artificial intelligence program. As a result of our study, we achieved the best results defined by artificial intelligence in the maxilla anterior region. There is a need for new studies in which bone destruction percentages can be measured by intraoral probing depth or calibrated radiographic data with CBCT. The limitations you stated have been added to the discussion and conclusion part.

In our study, 2 mm apical of the cemento-enamel junction was determined as the reference point in the radiography.

 Dear referee, we have mentioned above the articles that we received references.

Discussion

Point 7: Page 12, lines 388-390: This study simply assesses whether PBL is present or not. An extrapolation to AI guided construction of scaffolds for regeneration of periodontal defects is far beyond the published data. Delete paragraph.

Response 7: Dear reviewer, the aforementioned study was recommended by other reviewers who evaluated our study and added to the article. Since the results of our study are far beyond the published article, the paragraph has been removed from the text in line with your suggestion.

Point 8: Page 12, lines 411-414: This study simply assesses whether PBL is present or not. The authors repeatedly give the response that they did not assess severity of periodontal disease. However, now they conclude that AI “can provide a great convenience in detecting the stage, degree, severity, and localized/generalized or molar/incisor distribution of periodontal diseases”? This conclusion is also far beyond the published data. Delete sentence.

Response 8: This conclusion sentence also was deleted since far beyond the published data.

Data availability

Point 9: Page 13, lines 434: If the authors do not want to share open access data, they may state this clearly. The term “not applicable” is not appropriate.

Response 9: Thanks for the comment. We prefer not to share data because of the legal and ethical considerations in our state.

(Additions for revision 1 in our article are highlighted in yellow. Corrections and additions for the second revision are highlighted in green.)

Reviewer 2 Report

Authors have improved the manuscript significantly.

Author Response

Improvements and adjustments have been made in the sections you suggested. Additions for revision 1 in our article are highlighted in yellow. Corrections and additions for the second revision are highlighted in green.
